# Efficient Exploration in Continuous-time Model-based Reinforcement Learning

**Lenart Treven**
ETH Zürich
trevenl@ethz.ch

**Jonas Hübotter**
ETH Zürich
jhuebotter@student.ethz.ch

**Bhavya Sukhija**
ETH Zürich
sukhijab@ethz.ch

**Florian Dörfler**
ETH Zürich
dorfler@ethz.ch

**Andreas Krause**
ETH Zürich
krausea@ethz.ch

## Abstract

Reinforcement learning algorithms typically consider discrete-time dynamics, even though the underlying systems are often continuous in time. In this paper, we introduce a model-based reinforcement learning algorithm that represents continuous-time dynamics using nonlinear ordinary differential equations (ODEs). We capture epistemic uncertainty using well-calibrated probabilistic models, and use the optimistic principle for exploration. Our regret bounds surface the importance of the *measurement selection strategy* (MSS), since in continuous time we not only must decide how to explore, but also *when* to observe the underlying system. Our analysis demonstrates that the regret is sublinear when modeling ODEs with Gaussian Processes (GP) for common choices of MSS, such as equidistant sampling. Additionally, we propose an *adaptive*, data-dependent, practical MSS that, when combined with GP dynamics, also achieves sublinear regret with significantly fewer samples. We showcase the benefits of continuous-time modeling over its discrete-time counterpart, as well as our proposed *adaptive* MSS over standard baselines, on several applications.

## 1 Introduction

Real-world systems encountered in natural sciences and engineering applications, such as robotics (Spong et al., 2006), biology (Lenhart and Workman, 2007; Jones et al., 2009), medicine (Panetta and Fister, 2003), etc., are fundamentally continuous in time. Therefore, ordinary differential equations (ODEs) are the natural modeling language. However, the reinforcement learning (RL) community predominantly models problems in discrete time, with a few notable exceptions (Doya, 2000; Yildiz et al., 2021; Lutter et al., 2021). The discretization of continuous-time systems imposes limitations on the application of state-of-the-art RL algorithms, as they are tied to specific discretization schemes.

Discretization of continuous-time systems sacrifices several essential properties that could be preserved in a continuous-time framework (Nesic and Postoyan, 2021). For instance, exact discretization is only feasible for linear systems, leading to an inherent discrepancy when using standard discretization techniques for nonlinear systems. Additionally, discretization obscures the inter-sample behavior of the system, changes stability properties, may result in uncontrollable systems, or requires an excessively high sampling rate. Multi-time-scale systems are particularly vulnerable to these issues (Engquist et al., 2007). In many practical scenarios, the constraints imposed by discrete-time modeling are undesirable. Discrete-time models do not allow for the independent adjustment of measurement and control frequencies, which is crucial for real-world systems that operate in different regimes. For example, in autonomous driving, low-frequency sensor sampling

and control actuation may suffice at slow speeds, while high-speed driving demands faster control. How to choose aperiodic measurements and control is studied in the literature on the event and self-triggered control (Astrom and Bernhardsson, 2002; Anta and Tabuada, 2010; Heemels et al., 2012). They show that the number of control inputs can be significantly reduced by using aperiodic control. Moreover, data from multiple sources is often collected at varying frequencies (Ghysels et al., 2006) and is often not even equidistant in time. Discrete-time models struggle to exploit this data for learning, while continuous-time models can naturally accommodate it.

Continuous-time modeling also offers the flexibility to determine optimal measurement times based on need, in contrast to the fixed measurement frequency in discrete-time settings, which can easily miss informative samples. This advantage is particularly relevant in fields like medicine, where patient monitoring often requires higher frequency measurements during the onset of illness and lower frequency measurements as the patient recovers (Kaandorp and Koole, 2007). At the same time, in fields such as medicine, measurements are often costly, and hence, it is important that the most informative measurements are selected. Discrete-time models, limited to equidistant measurements, therefore often result in suboptimal decision-making and their sample efficiency is fundamentally limited. In summary, continuous-time modeling is agnostic to the choice of measurement selection strategy (MSS), whereas discrete-time modeling often only works for an equidistant MSS.

**Contributions**   Given the advantages of continuous-time modeling, in this work, we propose an optimistic continuous-time model-based RL algorithm – OCoRL. Moreover, we theoretically analyze OCoRL and show a general regret bound that holds for any MSS. We further show that for common choices of MSSs, such as equidistant sampling, the regret is sublinear when we model the dynamics with GPs (Williams and Rasmussen, 2006). To our knowledge, we are the first to give a no-regret algorithm for a rich class of nonlinear dynamical systems in the continuous-time RL setting. We further propose an *adaptive* MSS that is practical, data-dependent, and requires considerably fewer measurements compared to the equidistant MSS while still ensuring the sublinear regret for the GP case. Crucial to OCoRL is the exploration induced by the *optimism in the face of uncertainty* principle for model-based RL, which is commonly employed in the discrete-time realm (Auer et al., 2008; Curi et al., 2020). We validate the performance of OCoRL on several robotic tasks, where we clearly showcase the advantages of continuous-time modeling over its discrete-time counterpart. Finally, we provide an efficient implementation[1] of OCoRL in JAX (Bradbury et al., 2018).

## 2   Problem setting

In this work, we study a continuous-time deterministic dynamical system $\boldsymbol{f}^*$ with initial state $\boldsymbol{x}_0 \in \mathcal{X} \subset \mathbb{R}^{d_x}$, i.e.,

$$\boldsymbol{x}(t) = \boldsymbol{x}_0 + \int_0^t \boldsymbol{f}^* \left( \boldsymbol{x}(s), \boldsymbol{u}(s) \right) \, ds.$$

Here $\boldsymbol{u} : [0, \infty) \to \mathbb{R}^{d_u}$ is the input we apply to the system. Moreover, we consider state feedback controllers represented through a policy $\boldsymbol{\pi} : \mathcal{X} \to \mathcal{U} \subset \mathbb{R}^{d_u}$, that is, $\boldsymbol{u}(s) = \boldsymbol{\pi}(\boldsymbol{x}(s))$. Our objective is to find the optimal policy with respect to a given running cost function $c : \mathbb{R}^{d_x} \times \mathbb{R}^{d_u} \to \mathbb{R}$. Specifically, we are interested in solving the following optimal control (OC) problem over the policy space $\Pi$:

$$\boldsymbol{\pi}^* \overset{\text{def}}{=} \underset{\boldsymbol{\pi} \in \Pi}{\operatorname{argmin}} \, C(\boldsymbol{\pi}, \boldsymbol{f}^*) = \underset{\boldsymbol{\pi} \in \Pi}{\operatorname{argmin}} \int_0^T c(\boldsymbol{x}, \boldsymbol{\pi}(\boldsymbol{x})) \, dt \tag{1}$$
$$\text{s.t.} \quad \dot{\boldsymbol{x}} = \boldsymbol{f}^*(\boldsymbol{x}, \boldsymbol{\pi}(\boldsymbol{x})), \quad \boldsymbol{x}(0) = \boldsymbol{x}_0.$$

The function $\boldsymbol{f}^*$ is *unknown*, but we can collect data over episodes $n = 1 \ldots, N$ and learn about the system by deploying a policy $\boldsymbol{\pi}_n \in \Pi$ for the horizon of $T$ in episode $n$. In an (overly) idealized continuous time setting, one can measure the system at any time step. However, we consider a more practical setting, where we assume taking measurements is costly, and therefore want as few measurements as necessary. To this end, we formally define a measurement selection strategy below.

---

[1]`https://github.com/lenarttreven/ocorl`

**Definition 1** (Measurement selection strategy). *A measurement selection strategy $S$ is a sequence of sets $(S_n)_{n \geq 1}$, such that $S_n$ contains $m_n$ points at which we take measurements, i.e., $S_n \subset [0, T], |S_n| = m_n$.*[2]

During episode $n$, given a policy $\boldsymbol{\pi}_n$ and a MSS $S_n$, we collect a dataset $\mathcal{D}_n \sim (\boldsymbol{\pi}_n, S_n)$. The dataset is defined as

$$\mathcal{D}_n \stackrel{\text{def}}{=} \{(\boldsymbol{z}_n(t_{n,i}), \dot{\boldsymbol{y}}_n(t_{n,i})) \mid t_{n,i} \in S_n, i \in \{1 \ldots, m_n\}\} \qquad \text{where}$$

$$\boldsymbol{z}_n(t_{n,i}) \stackrel{\text{def}}{=} (\boldsymbol{x}_n(t_{n,i}), \boldsymbol{\pi}_n(\boldsymbol{x}_n(t_{n,i}))), \quad \dot{\boldsymbol{y}}_n(t_{n,i}) \stackrel{\text{def}}{=} \dot{\boldsymbol{x}}_n(t_{n,i}) + \boldsymbol{\epsilon}_{n,i}.$$

Here $\boldsymbol{x}_n(t), \dot{\boldsymbol{x}}_n(t)$ are state and state derivative in episode $n$, and $\boldsymbol{\epsilon}_{n,i}$ is i.i.d, $\sigma$-sub-Gaussian noise of the state derivative observations. Note, even though in practice only the state $\boldsymbol{x}(t)$ might be observable, one can estimate its derivative $\dot{\boldsymbol{x}}(t)$ (e.g., using finite differences, interpolation methods, etc. Cullum (1971); Knowles and Wallace (1995); Chartrand (2011); Knowles and Renka (2014); Wagner et al. (2018); Treven et al. (2021)). We capture the noise in our measurements and/or estimation of $\dot{\boldsymbol{x}}(t)$ with $\boldsymbol{\epsilon}_{i,n}$.

In summary, at each episode $n$, we deploy a policy $\boldsymbol{\pi}_n$ for a horizon of $T$, observe the system according to a proposed MSS $S_n$, and learn the dynamics $\boldsymbol{f}^*$. By deploying $\boldsymbol{\pi}_n$ instead of the optimal policy $\boldsymbol{\pi}^*$, we incur a regret,

$$r_n(S) \stackrel{\text{def}}{=} C(\boldsymbol{\pi}_n, \boldsymbol{f}^*) - C(\boldsymbol{\pi}^*, \boldsymbol{f}^*).$$

Note that the policy $\boldsymbol{\pi}_n$ depends on the data $\mathcal{D}_{1:n-1} = \cup_{i<n} \mathcal{D}_i$ and hence implicitly on the MSS $S$.

**Performance measure** We analyze OCoRL by comparing it with the performance of the best policy $\boldsymbol{\pi}^*$ from the class $\Pi$. We evaluate the *cumulative regret* $R_N(S) \stackrel{\text{def}}{=} \sum_{n=1}^{N} r_n(S)$ that sums the gaps between the performance of the policy $\boldsymbol{\pi}_n$ and the optimal policy $\boldsymbol{\pi}^*$ over all the episodes. If the regret $R_N(S)$ is sublinear in $N$, then the average cost of the policy $C(\boldsymbol{\pi}_n, \boldsymbol{f}^*)$ converges to the optimal cost $C(\boldsymbol{\pi}^*, \boldsymbol{f}^*)$.

## 2.1 Assumptions

Any meaningful analysis of cumulative regret for continuous time, state, and action spaces requires some assumptions on the system and the policy class. We make some continuity assumptions, similar to the discrete-time case (Khalil, 2015; Curi et al., 2020; Sussex et al., 2023), on the dynamics, policy, and cost.

**Assumption 1** (Lipschitz continuity). *Given any norm $\|\cdot\|$, we assume that the system dynamics $\boldsymbol{f}^*$ and cost $c$ are $L_{\boldsymbol{f}}$ and $L_c$-Lipschitz continuous, respectively, with respect to the induced metric. Moreover, we define $\Pi$ to be the policy class of $L_{\boldsymbol{\pi}}$-Lipschitz continuous policy functions and $\mathcal{F}$ a class of $L_{\boldsymbol{f}}$ Lipschitz continuous dynamics functions with respect to the induced metric.*

We learn a model of $\boldsymbol{f}^*$ using data collected from the episodes. For a given state-action pair $\boldsymbol{z} = (\boldsymbol{x}, \boldsymbol{u})$, our learned model predicts a mean estimate $\boldsymbol{\mu}_n(\boldsymbol{z})$ and quantifies our epistemic uncertainty $\boldsymbol{\sigma}_n(\boldsymbol{z})$ about the function $\boldsymbol{f}^*$.

**Definition 2** (Well-calibrated statistical model of $\boldsymbol{f}^*$, Rothfuss et al. (2023)). *Let $\mathcal{Z} \stackrel{\text{def}}{=} \mathcal{X} \times \mathcal{U}$. An all-time well-calibrated statistical model of the function $\boldsymbol{f}^*$ is a sequence $\{\mathcal{M}_n(\delta)\}_{n \geq 0}$, where*

$$\mathcal{M}_n(\delta) \stackrel{\text{def}}{=} \left\{ \boldsymbol{f} : \mathcal{Z} \to \mathbb{R}^{d_x} \mid \forall \boldsymbol{z} \in \mathcal{Z}, \forall j \in \{1, \ldots, d_x\} : |\mu_{n,j}(\boldsymbol{z}) - f_j(\boldsymbol{z})| \leq \beta_n(\delta)\sigma_{n,j}(\boldsymbol{z}) \right\},$$

*if, with probability at least $1 - \delta$, we have $\boldsymbol{f}^* \in \bigcap_{n \geq 0} \mathcal{M}_n(\delta)$. Here, $\mu_{n,j}$ and $\sigma_{n,j}$ denote the $j$-th element in the vector-valued mean and standard deviation functions $\boldsymbol{\mu}_n$ and $\boldsymbol{\sigma}_n$ respectively, and $\beta_n(\delta) \in \mathbb{R}_{\geq 0}$ is a scalar function that depends on the confidence level $\delta \in (0, 1]$ and which is monotonically increasing in $n$.*

**Assumption 2** (Well-calibration). *We assume that our learned model is an all-time well-calibrated statistical model of $\boldsymbol{f}^*$. We further assume that the standard deviation functions $(\boldsymbol{\sigma}_n(\cdot))_{n \geq 0}$ are $L_{\boldsymbol{\sigma}}$-Lipschitz continuous.*

---

[2]Here, the set $S_n$ may depend on observations prior to episode $n$ or is even constructed while we execute the trajectory. For ease of notation, we do not make this dependence explicit.

This is a natural assumption, which states that we are with high probability able to capture the dynamics within a confidence set spanned by our predicted mean and epistemic uncertainty. For example, GP models are all-time well-calibrated for a rich class of functions (c.f., Section 3.2) and also satisfy the Lipschitz continuity assumption on $(\boldsymbol{\sigma}_n(\cdot))_{n \geq 0}$ (Rothfuss et al., 2023). For Bayesian neural networks, obtaining accurate uncertainty estimates is still an open and active research problem. However, in practice, re-calibration techniques (Kuleshov et al., 2018) can be used.

By leveraging these assumptions, in the next section, we propose our algorithm OCoRL and derive a generic bound on its cumulative regret. Furthermore, we show that OCoRL provides sublinear cumulative regret for the case when GPs are used to learn $\boldsymbol{f}^*$.

# 3   Optimistic Continuous-time RL Algorithm

---

**OCoRL:** OPTIMISTIC CONTINUOUS-TIME RL

    **Init:** Statistical model $\mathcal{M}_0$, Simulator SIM, MSS $S$, Probability $\delta$
    **for** episode $n = 1, \ldots, N$ **do**

$$\boldsymbol{\pi}_n = \underset{\boldsymbol{\pi} \in \Pi}{\operatorname{argmin}} \ \underset{\boldsymbol{f} \in \mathcal{M}_{n-1} \cap \mathcal{F}}{\min} C(\boldsymbol{\pi}, \boldsymbol{f}) \qquad \text{\color{blue}/* Select optimistic policy /*}$$

$$\mathcal{D}_n = \{(\boldsymbol{z}_n(t_{n,i}), \dot{\boldsymbol{y}}_n(t_{n,i})) \mid t_{n,i} \in S_n\} \leftarrow \text{SIM}(\boldsymbol{\pi}_n, S_n) \qquad \text{\color{blue}/* Measurement collection /*}$$

$$\mathcal{M}_n \leftarrow (\boldsymbol{\mu}_n, \boldsymbol{\sigma}_n, \beta_n(\delta)) \leftarrow \mathcal{D}_{1:n} \qquad \text{\color{blue}/* Update statistical model/*}$$

---

**Optimistic policy selection**   There are several strategies we can deploy to trade-off exploration and exploitation, e.g., dithering ($\varepsilon$-greedy, Boltzmann exploration (Sutton and Barto, 2018)), Thompson sampling (Osband et al., 2013), upper-confidence RL (UCRL) (Auer et al., 2008), etc. OCoRL is a continuous-time variant of the Hallucinated UCRL strategy introduced by Chowdhury and Gopalan (2017); Curi et al. (2020). In episode $n$, the optimistic policy is obtained by solving the optimal control problem:

$$(\boldsymbol{\pi}_n, \boldsymbol{f}_n) \overset{\text{def}}{=} \underset{\boldsymbol{\pi} \in \Pi, \ \boldsymbol{f} \in \mathcal{M}_{n-1} \cap \mathcal{F}}{\operatorname{argmin}} C(\boldsymbol{\pi}, \boldsymbol{f}) \tag{2}$$

Here, $\boldsymbol{f}_n$ is a dynamical system such that the cost by controlling $\boldsymbol{f}_n$ with its optimal policy $\boldsymbol{\pi}_n$ is the lowest among all the plausible systems from $\mathcal{M}_{n-1} \cap \mathcal{F}$. The optimal control problem (2) is infinite-dimensional, in general nonlinear, and thus hard to solve. For the analysis, we assume we can perfectly solve it. In Appendix B, we present details on how we solve it in practice. Specifically, in Appendix B.1, we show that our results seamlessly extend to the setting where the optimal control problem of Equation (1) is discretized w.r.t. an *arbitrary* choice of discretization. Moreover, in this setting, we show that theoretical guarantees can be derived without restricting the models to $\boldsymbol{f} \in \mathcal{M}_{n-1} \cap \mathcal{F}$. Instead, a practical optimization over all models in $\mathcal{M}_{n-1}$ can be performed as in Curi et al. (2020); Pasztor et al. (2021).

**Model complexity**   We expect that the regret of any model-based continuous-time RL algorithm depends both on the hardness of learning the underlying true dynamics model $\boldsymbol{f}^*$ and the MSS. To capture both, we define the following model complexity:

$$\mathcal{I}_N(\boldsymbol{f}^*, S) \overset{\text{def}}{=} \underset{\substack{\boldsymbol{\pi}_1, \ldots, \boldsymbol{\pi}_N \\ \boldsymbol{\pi}_n \in \Pi}}{\max} \sum_{n=1}^{N} \int_0^T \left\| \boldsymbol{\sigma}_{n-1}(\boldsymbol{z}_n(t)) \right\|^2 \, dt. \tag{3}$$

The model complexity measure captures the hardness of learning the dynamics $\boldsymbol{f}^*$. Intuitively, for a given $N$, the more complicated the dynamics $\boldsymbol{f}^*$, the larger the epistemic uncertainty and thereby the model complexity. For the discrete-time setting, the integral is replaced by the sum over the uncertainties on the trajectories (c.f., Equation (8) of Curi et al. (2020)). In the continuous-time setting, we do not observe the state at every time step, but only at a finite number of times wherever the MSS $S$ proposes to measure the system. Accordingly, $S$ influences how we collect data and update our calibrated model. Therefore, the model complexity depends on $S$. Next, we first present the regret bound for general MSSs, then we look at particular strategies for which we can show convergence of the regret.

**Proposition 1.** *Let $S$ be any MSS. If we run OCORL, we have with probability at least $1 - \delta$,*

$$R_N(S) \leq 2\beta_N L_c (1 + L_{\boldsymbol{\pi}}) T^{\frac{3}{2}} e^{L_{\boldsymbol{f}}(1+L_{\boldsymbol{\pi}})T} \sqrt{N \mathcal{I}_N(\boldsymbol{f}^*, S)}. \tag{4}$$

We provide the proof of Proposition 1 in Appendix A. Because we have access only to the statistical model and any errors in dynamics compound (continuously in time) over the episode, the regret $R_N(S)$ depends exponentially on the horizon $T$. This is in line with the prior work in the discrete-time setting (Curi et al., 2020). If the model complexity term $\mathcal{I}_N(\boldsymbol{f}^*, S)$ and $\beta_N$ grow at a rate slower than $N$, the regret is sublinear and the average performance of OCORL converges to $C(\boldsymbol{\pi}^*, \boldsymbol{f}^*)$. In our analysis, the key step to show sublinear regret for the GP dynamics model is to upper bound the integral of uncertainty in the model complexity with the sum of uncertainties at the points where we collect the measurements. In the next section, we show how this can be done for different measurement selection strategies.

### 3.1 Measurement selection strategies (MSS)

In the following, we present different natural MSSs and compare the number of measurements they propose per episode. Formal derivations are included in Appendix A.2.

**Oracle** Intuitively, if we take measurements at the times when we are the most uncertain about dynamics on the executed trajectory, i.e., when $\|\boldsymbol{\sigma}_{n-1}(\boldsymbol{z}_n(t))\|$ is largest, we gain the most knowledge about the true function $\boldsymbol{f}^*$. Indeed, when the statistical model is a GP and noise is homoscedastic and Gaussian, observing the most uncertain point on the trajectory leads to the maximal reduction in entropy of $\boldsymbol{f}^*(\boldsymbol{z}_n(t))$ (c.f., Lemma 5.3 of Srinivas et al. (2009)). In the ideal case, we can define an MSS that collects only the point with the highest uncertainty in every episode, i.e., $S_n^{\text{ORA}} \stackrel{\text{def}}{=} \{t_{n,1}\}$ where $t_{n,1} \stackrel{\text{def}}{=} \text{argmax}_{0 \leq t < T} \|\boldsymbol{\sigma}_{n-1}(\boldsymbol{z}_n(t))\|^2$. For this MSS we bound the integral over the horizon $T$ with the maximal value of the integrand times the horizon $T$:

$$\mathcal{I}_N(\boldsymbol{f}^*, S_n^{\text{ORA}}) \leq \max_{\substack{\boldsymbol{\pi}_1, \ldots, \boldsymbol{\pi}_N \\ \boldsymbol{\pi}_n \in \Pi}} T \sum_{n=1}^N \|\boldsymbol{\sigma}_{n-1}(\boldsymbol{z}_n(t_{n,1}))\|^2 \tag{5}$$

The *oracle* MSS collects only one measurement per episode, however, it is impractical since it requires knowing the most uncertain point on the true trajectory a priori, i.e., before executing the policy.

**Equidistant** Another natural MSS is the equidistant MSS. We collect $m_n$ equidistant measurements in episode $n$ and upper bound the integral with the upper Darboux integral.

$$\mathcal{I}_N(\boldsymbol{f}^*, S_n^{\text{EQI}}) \leq \max_{\substack{\boldsymbol{\pi}_1, \ldots, \boldsymbol{\pi}_N \\ \boldsymbol{\pi}_n \in \Pi}} \sum_{n=1}^N \frac{T}{m_n} \sum_{i=1}^{m_n} \left( \|\boldsymbol{\sigma}_{n-1}(\boldsymbol{z}_n(t_{n,i}))\|^2 + \frac{T L_{\boldsymbol{\sigma}^2}}{m_n} \right). \tag{6}$$

Here $L_{\boldsymbol{\sigma}^2}$ is the Lipschitz constant of $\|\boldsymbol{\sigma}_{n-1}(\cdot)\|^2$. To achieve sublinear regret, we require that $\sum_{n=1}^N \frac{1}{m_n} \in o(N)$. Therefore, for a fixed equidistant MSS, our analysis does not ensure a sublinear regret. This is because we consider a continuous-time regret which is integrated (c.f., Equation (1)) while in the discrete-time setting, the regret is defined for the equidistant MSS only (Curi et al., 2020). Accordingly, we study a strictly harder problem. Nonetheless, by linearly increasing the number of observations per episode and setting $m_n = n$, we get $\sum_{n=1}^N \frac{1}{m_n} \in \mathcal{O}(\log(N))$ and sublinear regret. The equidistant MSS is easy to implement, however, the required number of samples is increasing linearly with the number of episodes.

**Adaptive** Finally, we present an MSS that is practical, i.e., easy to implement and at the same time requires only a few measurements per episode. The core idea of receding horizon adaptive MSS is simple: simulate (hallucinate) the system $\boldsymbol{f}_n$ with the policy $\boldsymbol{\pi}_n$, and find the time $t$ such that $\|\boldsymbol{\sigma}_{n-1}(\widehat{\boldsymbol{z}}_n(t))\|$ is largest. Here, $\widehat{\boldsymbol{z}}_n(t)$ is the state-action pair at time $t$ in episode $n$ for the hallucinated trajectory.

However, the hallucinated trajectory can deviate from the true trajectory exponentially fast in time, and the time of the largest variance on the hallucinated trajectory can be far away from the time of the largest variance on the true trajectory. To remedy this technicality, we utilize a receding horizon MSS. We split, in episode $n$, the time horizon $T$ into uniform-length intervals of $\Delta_n$ time, where $\Delta_n \in \Omega(\frac{1}{\beta_n})$, c.f., Appendix A. At the beginning of every time interval, we measure the true state-action pair $z$, hallucinate with policy $\pi_n$ starting from $z$ for $\Delta_n$ time on the system $f_n$, and calculate the time when the hallucinated trajectory has the highest variance. Over the next time horizon $\Delta_n$, we collect a measurement only at that time. Formally, let $m_n = \lceil T/\Delta_n \rceil$ be the number of measurements in episode $n$ and denote for every $i \in \{1, \dots, m_n\}$ the time with the highest variance on the hallucinated trajectory in the time interval $[(i-1)\Delta_n, i\Delta_n]$ by $t_{n,i}$:

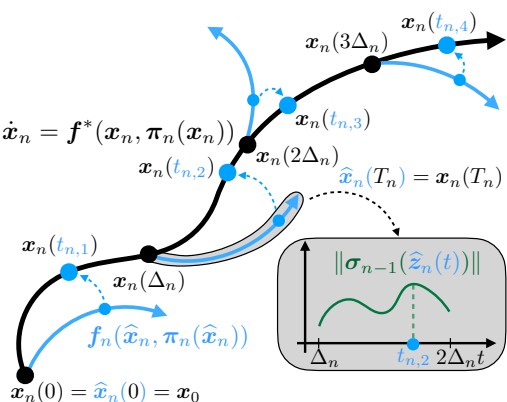

Figure 1: In episode $n$ we split the horizon $T$ into intervals of $\Delta_n$ time. We hallucinate the trajectory in every interval and select time $t_{n,i}$ in the interval $i$ where the uncertainty on the hallucinated trajectory is the highest.

$$t_{n,i} \stackrel{\text{def}}{=} (i-1)\Delta_n + \underset{0 \leq t \leq \Delta_n}{\operatorname{argmax}} \|\boldsymbol{\sigma}_{n-1}(\widehat{\boldsymbol{z}}_n(t))\|$$

$$\text{s.t.} \quad \dot{\widehat{\boldsymbol{x}}}_n = \boldsymbol{f}_n(\widehat{\boldsymbol{x}}_n, \boldsymbol{\pi}_n(\widehat{\boldsymbol{x}}_n))$$

$$\widehat{\boldsymbol{x}}_n(0) = \boldsymbol{x}_n((i-1)\Delta_n)$$

For this MSS, we show in Appendix A the upper bound

$$\mathcal{I}_N(\boldsymbol{f}^*, S_n^{\text{ADP}}) \leq \max_{\substack{\boldsymbol{\pi}_1, \dots, \boldsymbol{\pi}_N \\ \boldsymbol{\pi}_n \in \Pi}} 9 \sum_{n=1}^{N} \frac{T}{m_n} \sum_{i=1}^{m_n} \|\boldsymbol{\sigma}_{n-1}(\boldsymbol{z}_n(t_{n,i}))\|^2. \tag{7}$$

The model complexity upper bound for the adaptive MSS in Equation (7) does not have the $L_{\boldsymbol{\sigma}^2}$-dependent term from the equidistant MSS (6) and only depends on the sum of uncertainties at the collected points. Further, the number of points we collect in episode $n$ is $\mathcal{O}(\beta_n)$, and $\beta_n$, e.g., if we model the dynamics with GPs with radial basis function (RBF) kernel, is of order $\text{polylog}(n)$ (c.f., Lemma 2).

### 3.2 Modeling dynamics with a Gaussian Process

We can prove sublinear regret for the proposed three MSSs coupled with optimistic exploration when we model the dynamics using GPs with standard kernels such as RBF, linear, etc. We consider the vector-valued dynamics model $\boldsymbol{f}^*(\boldsymbol{z}) = (f_1^*(\boldsymbol{z}), \dots, f_{d_x}^*(\boldsymbol{z}))$, where scalar-valued functions $f_j^* \in \mathcal{H}_k$ are elements of a Reproducing Kernel Hilbert Space (RKHS) $\mathcal{H}_k$ with kernel function $k$, and their norm is bounded $\|f_j^*\|_k \leq B$. We write $\boldsymbol{f}^* \in \mathcal{H}_{k,B}^{d_x} \stackrel{\text{def}}{=} \{(f_1, \dots, f_{d_x}) \mid \|f_j\|_k \leq B\}$.

To learn $\boldsymbol{f}^*$ we fit a GP model with a zero mean and kernel $k$ to the collected data $\mathcal{D}_{1:n}$. For ease of notation, we denote by $\dot{\boldsymbol{y}}_{1:n}^j$ the concatenated $j$-th dimension of the state derivative observations from $\mathcal{D}_{1:n}$. The posterior means and variances of the GP model have a closed form (Williams and Rasmussen (2006)):

$$\mu_{n,j}(\boldsymbol{z}) = \boldsymbol{k}_n^\top(\boldsymbol{z}) \left(\boldsymbol{K}_n + \sigma^2 \boldsymbol{I}\right)^{-1} \dot{\boldsymbol{y}}_{1:n}^j$$

$$\sigma_{n,j}^2(\boldsymbol{z}) = k(\boldsymbol{z}, \boldsymbol{z}) - \boldsymbol{k}_n^\top(\boldsymbol{z}) \left(\boldsymbol{K}_n + \sigma^2 \boldsymbol{I}\right)^{-1} \boldsymbol{k}_n(\boldsymbol{z}),$$

where we write $\boldsymbol{K}_n = [k(\boldsymbol{z}_l, \boldsymbol{z}_m)]_{\boldsymbol{z}_l, \boldsymbol{z}_m \in \mathcal{D}_{1:n}}$, and $\boldsymbol{k}_n(\boldsymbol{z}) = [k(\boldsymbol{z}_l, \boldsymbol{z})]_{\boldsymbol{z}_l \in \mathcal{D}_{1:n}}$. The posterior means and variances with the right scaling factor $\beta_n(\delta)$ satisfy the all-time well-calibrated Assumption 2.

**Lemma 2** (Lemma 3.6 from Rothfuss et al. (2023)). *Let* $\boldsymbol{f}^* \in \mathcal{H}_{k,B}^{d_x}$ *and* $\delta \in (0, 1)$. *Then there exist* $\beta_n(\delta) \in \mathcal{O}(\sqrt{\gamma_n + \log(1/\delta)})$ *such that the confidence sets* $\mathcal{M}_n$ *built from the triplets* $(\boldsymbol{\mu}_n, \boldsymbol{\sigma}_n, \beta_n(\delta))$ *form an all-time well-calibrated statistical model.*

Here, $\gamma_n$ is the *maximum information gain* after observing $n$ points (Srinivas et al., 2009), as defined in Appendix A, where we also provide the rates for common kernels. For example, for the RBF kernel, $\gamma_n = \mathcal{O}\left(\log(n)^{d_x+d_u+1}\right)$.

Finally, we show sublinear regret for the proposed MSSs for the case when we model dynamics with GPs. The proof of Theorem 3 is provided in Appendix A.

**Theorem 3.** *Assume that $\boldsymbol{f}^* \in \mathcal{H}_{k,B}^{d_x}$, the observation noise is i.i.d. $\mathcal{N}(\boldsymbol{0}; \sigma^2 \boldsymbol{I})$, and $\|\cdot\|$ is the Euclidean norm. We model $\boldsymbol{f}^*$ with the GP model. The regret for different MSSs is with probability at least $1 - \delta$ bounded by*

$$R_N(S^{ORA}) \leq \mathcal{O}\left(\beta_N T^2 e^{L_f(1+L_\pi)T} \sqrt{N\gamma_N}\right), \qquad\qquad m_n^{ORA} = 1$$

$$R_N(S^{EQI}) \leq \mathcal{O}\left(\beta_N T^2 e^{L_f(1+L_\pi)T} \sqrt{N\left(\gamma_N + \log(N)\right)}\right), \qquad m_n^{EQI} = n$$

$$R_N(S^{ADP}) \leq \mathcal{O}\left(\beta_N T^2 e^{L_f(1+L_\pi)T} \sqrt{N\gamma_N}\right), \qquad\qquad m_n^{ADP} = \mathcal{O}(\beta_n)$$

The optimistic exploration coupled with any of the proposed MSSs achieves regret that depends on the maximum information gain. All regret bounds from Theorem 3 are sublinear for common kernels, like linear and RBF. The bound for the adaptive MSS with RBF kernel is $\mathcal{O}\left(\sqrt{N\log(N)^{2(d_x+d_u+1)}}\right)$, where we hide the dependence on $T$ in the $\mathcal{O}$ notation. We reiterate that while the number of observations per episode of the oracle MSS is optimal, we cannot implement the oracle MSS in practice. In contrast, the equidistant MSS is easy to implement, but the number of measurements grows linearly with episodes. Finally, adaptive MSS is practical, i.e., easy to implement, and requires only a few measurements per episode, i.e., polylog($n$) in episode $n$ for RBF.

**Summary**   OCoRL consists of two key and orthogonal components; *(i)* optimistic policy selection and *(ii)* measurement selection strategies (MSSs). In optimistic policy selection, we optimistically, w.r.t. plausible dynamics, plan a trajectory and rollout the resulting policy. We study different MSSs, such as the typical equidistant MSS, for data collection within the framework of continuous time modeling. Furthermore, we propose an adaptive MSS that measures data where we have the highest uncertainty on the planned (hallucinated) trajectory. We show that OCoRL suffers no regret for the equidistant, adaptive, and oracle MSSs.

## 4   Related work

**Model-based Reinforcement Learning**   Model-based reinforcement learning (MBRL) has been an active area of research in recent years, addressing the challenges of learning and exploiting environment dynamics for improved decision-making. Among the seminal contributions, Deisenroth and Rasmussen (2011) proposed the PILCO algorithm which uses Gaussian processes for learning the system dynamics and policy optimization. Chua et al. (2018) used deep ensembles as dynamics models. They coupled MPC (Morari and Lee, 1999) to efficiently solve high dimensional tasks with considerably better sample complexity compared to the state-of-the-art model-free methods SAC (Haarnoja et al., 2018), PPO (Schulman et al., 2017), and DDPG (Lillicrap et al., 2015). The aforementioned model-based methods use more or less greedy exploitation that is provably optimal only in the case of the linear dynamics (Simchowitz and Foster, 2020). Exploration methods based on Thompson sampling (Dearden et al., 2013; Chowdhury and Gopalan, 2019) and Optimism (Auer et al., 2008; Abbasi-Yadkori and Szepesvári, 2011; Luo et al., 2018; Curi et al., 2020), however, provably converge to the optimal policy also for a large class of nonlinear dynamics if modeled with GPs. Among the discrete-time RL algorithms, our work is most similar to the work of Curi et al. (2020) where they use the reparametrization trick to explore optimistically among all statistically plausible discrete dynamical models.

**Continuous-time Reinforcement Learning**   Reinforcement learning in continuous-time has been around for several decades (Doya, 2000; Vrabie and Lewis, 2008, 2009; Vamvoudakis et al., 2009). Recently, the field has gained more traction following the work on Neural ODEs by Chen et al. (2018). While physics biases such as Lagrangian (Cranmer et al., 2020) or Hamiltonian (Greydanus et al., 2019) mechanics can be enforced in continuous-time modeling, different challenges such

as vanishing $Q$-function (Tallec et al., 2019) need to be addressed in the continuous-time setting. Yildiz et al. (2021) introduce a practical episodic model-based algorithm in continuous time. In each episode, they use the learned mean estimate of the ODE model to solve the optimal control task with a variant of a continuous-time actor-critic algorithm. Compared to Yildiz et al. (2021) we solve the optimal control task using the optimistic principle. Moreover, we thoroughly motivate optimistic planning from a theoretical standpoint. Lutter et al. (2021) introduce a continuous fitted value iteration and further show a successful hardware application of their continuous-time algorithm. A vast literature exists for continuous-time RL with linear systems (Modares and Lewis, 2014; Wang et al., 2020), but few, only for linear systems, provide theoretical guarantees (Mohammadi et al., 2021; Basei et al., 2022). To the best of our knowledge, we are the first to provide theoretical guarantees of *convergence to the optimal cost in continuous time* for a large class of RKHS functions.

**Aperiodic strategies**   Aperiodic MSSs and controls (event and self-triggered control) are mostly neglected in RL since most RL works predominantly focus on discrete-time modeling. There exists a considerable amount of literature on the event and self-triggered control (Astrom and Bernhardsson, 2002; Anta and Tabuada, 2010; Heemels et al., 2012). However, compared to periodic control, its theory is far from being mature (Heemels et al., 2021). In our work, we assume that we can control the system continuously, and rather focus on when to measure the system instead. The closest to our adaptive MSS is the work of Du et al. (2020), where they empirically show that by optimizing the number of interaction times, they can achieve similar performance (in terms of cost) but with fewer interactions. Compared to us, they do not provide any theoretical guarantees. Umlauft and Hirche (2019); Lederer et al. (2021) consider the non-episodic setting where they can continuously monitor the system. They suggest taking measurements only when the uncertainty of the learned model on the monitored trajectory surpasses the boundary ensuring stability. They empirically show, for feedback linearizable systems, that by applying their strategy the number of measurements reduces drastically and the tracking error remains bounded. Compared to them, we consider general dynamical systems and also don't assume continuous system monitoring.

## 5   Experiments

We now empirically evaluate the performance of OCoRL on several environments. We test OCoRL on *Cancer Treatment and Glucose in blood systems* from Howe et al. (2022), *Pendulum*, *Mountain Car* and *Cart Pole* from Brockman et al. (2016), *Bicycle* from Polack et al. (2017), *Furuta Pendulum* from Lutter et al. (2021) and *Quadrotor in 2D and 3D* from Nonami et al. (2010). The details of the systems' dynamics and tasks are provided in Appendix C.

**Comparison methods**   To make the comparison fair, we adjust methods so that they all collect the same number of measurements per episode. For the equidistant setting, we collect $M$ points per episode (we provide values of $M$ for different systems in Appendix C). For the adaptive MSS, we assume $\Delta_n \geq T$, and instead of one measurement per episode we collect a batch of $M$ measurements such that they (as a batch) maximize the variance on the hallucinated trajectory. To this end, we consider the *Greedy Max Determinant* and *Greedy Max Kernel Distance* strategies of Holzmüller et al. (2022). We provide details of the adaptive strategies in Appendix C. We compare OCoRL with the described MSSs to the optimal discrete-time zero hold control, where we assume the access to the true discretized dynamics $\boldsymbol{f}_d^*(\boldsymbol{x}, \boldsymbol{u}) = \boldsymbol{x} + \int_0^{T/(M-1)} \boldsymbol{f}^*(\boldsymbol{x}(t), \boldsymbol{u}) \, dt$. We further also compare with the best continuous-time control policy, i.e., the solution of Equation (1).

**Does the continuous-time control policy perform better than the discrete-time control policy?** In the first experiment, we test whether learning a continuous-time model from the finite data coupled with a continuous-time control policy on the learned model can outperform the discrete-time zero-order hold control on the true system. We conduct the experiment on all environments and report the cost after running OCoRL for a few tens of episodes (the exact experimental details are provided in Appendix C). From Table 1, we conclude that the OCoRL outperforms the discrete-time zero-order hold control on the true model on every system if we use the adaptive MSS, while achieving lower cost on 7 out of 9 systems if we measure the system equidistantly.

Table 1: OCoRL with adaptive MSSs achieves lower final cost $C(\boldsymbol{\pi}_N, \boldsymbol{f}^*)$ compared to the discrete-time control on the true system on all tested environments while converging towards the best continuous-time control policy. While equidistant MSS achieves higher cost compared to the adaptive MSS, it still outperforms the discrete-time zero-order hold control on the true model for most systems.

| | Known True Model | | Optimistic Exploration with different MSS | | |
|---|---|---|---|---|---|
| System | Continuous time OC | Discrete zero-order hold OC | Max Kernel Distance | Max Determinant | Equidistant |
| Cancer Treatment | 20.57 | 21.05 | $20.70 \pm 0.06$ | $\mathbf{20.68 \pm 0.05}$ | $21.05 \pm 1.60$ |
| Glucose in Blood | 15.23 | 15.30 | $\mathbf{15.23 \pm 0.01}$ | $15.24 \pm 0.01$ | $15.25 \pm 0.01$ |
| Pendulum | 20.16 | 20.59 | $\mathbf{20.20 \pm 0.02}$ | $\mathbf{20.20 \pm 0.02}$ | $20.29 \pm 0.03$ |
| Mountain Car | 34.63 | 35.04 | $\mathbf{34.63 \pm 0.01}$ | $\mathbf{34.63 \pm 0.01}$ | $34.64 \pm 0.01$ |
| Cart Pole | 17.49 | 19.96 | $\mathbf{17.52 \pm 0.04}$ | $17.53 \pm 0.04$ | $17.63 \pm 0.05$ |
| Bicycle | 9.45 | 10.24 | $\mathbf{9.53 \pm 0.02}$ | $9.53 \pm 0.03$ | $9.67 \pm 0.05$ |
| Furuta Pendulum | 23.31 | 25.11 | $23.64 \pm 0.22$ | $\mathbf{23.52 \pm 0.18}$ | $314.77 \pm 411.01$ |
| Quadrotor 2D | 3.54 | 4.01 | $\mathbf{3.54 \pm 0.01}$ | $\mathbf{3.54 \pm 0.01}$ | $3.57 \pm 0.01$ |
| Quadrotor 3D | 7.38 | 7.84 | $\mathbf{7.51 \pm 0.21}$ | $7.54 \pm 0.28$ | $9.41 \pm 1.43$ |

**Does the adaptive MSS perform better than equidistant MSS?** We compare the adaptive and equidistant MSSs on all systems and observe that the adaptive MSSs consistently perform better than the equidistant MSS. To better illustrate the difference between the adaptive and equidistant MSSs we study a 2-dimensional Pendulum system (c.f., Figure 2). First, we see that if we use adaptive MSSs we consistently achieve lower per-episode costs during the training. Second, we observe that while equidistant MSS spreads observations equidistantly in time and collects lots of measurements with almost the same state-action input of the dynamical system, the adaptive MSS spreads the measurements to have diverse state-action input pairs for the dynamical system on the executed trajectory and collects higher quality data.

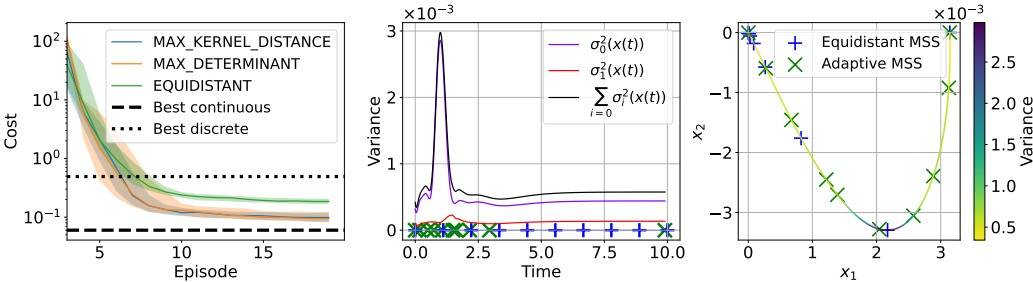

Figure 2: All MSSs coupled with continuous-time control achieve lower cost than the optimal discrete zero-order hold control on the true model. Adaptive MSSs (Greedy Max Kernel Distance and Greedy Max Determinant) reduce the suffered cost considerably faster than equidistant MSS and are converging towards the best possible continuous-time control. Whereas equidistant MSS spreads the measurements uniformly over time, the adaptive MSSs spread the data over the state-action space (dynamical system's input) and collect higher quality data.

**Does optimism help?** For the final experiment, we examine whether planning optimistically helps. In particular, we compare our planning strategy to the mean planner that is also used by Yildiz et al. (2021). The mean planner solves the optimal control problem greedily with the learned mean model $\boldsymbol{\mu}_n$ in every episode $n$. We evaluate the performance of this model on the Pendulum system for all MSSs. We observe that planning optimistically results in reducing the cost faster and achieves better performance for all MSSs.

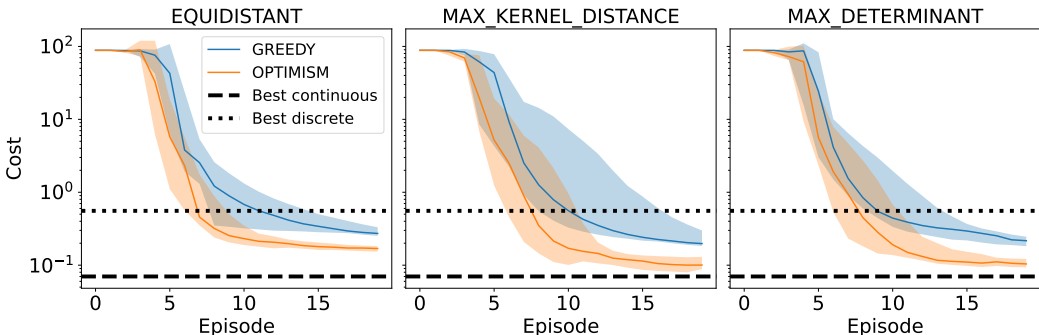

Figure 3: Both greedy and optimistic continuous-time control strategies outperform the discrete-time zero-order hold control on the true model. However, the optimistic strategy is particularly notable as it expedites cost reduction across all MSSs.

## 6 Conclusion

We introduce a model-based continuous-time RL algorithm OCoRL that uses the *optimistic paradigm* to provably achieve sublinear regret, with respect to the best possible continuous-time performance, for several MSSs if modeled with GPs. Further, we develop a practical *adaptive* MSS that, compared to the standard equidistant MSS, drastically reduces the number of measurements per episode while retaining the regret guarantees. Finally, we showcase the benefits of continuous-time compared to discrete-time modeling in several environments (c.f. Figure 1, Table 1), and demonstrate the benefits of planning with *optimism* compared to greedy planning.

In this work, we considered the setting with deterministic dynamics where we obtain a noisy measurement of the state's derivative. We leave the more practical setting, where only noisy measurements of the state, instead of its derivative, are available as well as stochastic, delayed differential equations, and partially observable systems to future work.

Our aim with this work is to catalyze further research within the RL community on continuous-time modeling. We believe that this shift in perspective could lead to significant advancements in the field and look forward to future contributions.

## Acknowledgments and Disclosure of Funding

We would like to thank Yarden As, Scott Sussex and Dominik Baumann for their feedback. This project has received funding from the Swiss National Science Foundation under NCCR Automation, grant agreement 51NF40 180545, and the Microsoft Swiss Joint Research Center.

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

# Contents of Appendix

# A  Theory

We begin in Appendix A.1 by deriving the cumulative regret bound of Proposition 1 which depends on the model complexity. In Appendix A.2, we bound the model complexities of the presented MSSs. Finally, in Appendix A.3, we bound the measurement uncertainty when a GP is used as the statistical model, yielding sublinear regret guarantees for many commonly used kernels. We include useful facts and inequalities in Appendix A.4.

## A.1   Bounding Regret with the Model Complexity

The main goal of this section is to derive Proposition 1 which relates the cumulative regret $R_N(S)$ to the model complexity $\mathcal{I}_N(\boldsymbol{f}^*, S)$. In what follows, the quantifier "with high probability" refers to "with probability at least $1 - \delta$ as in Definition 2".

**Lemma 4.** *Assuming $\boldsymbol{x}_n(0) = \widehat{\boldsymbol{x}}_n(0)$, the distance between the true and the hallucinated trajectory at any time $t \geq 0$ is bounded with high probability by*

$$\|\widehat{\boldsymbol{x}}_n(t) - \boldsymbol{x}_n(t)\| \leq 2\beta_n e^{L_{\boldsymbol{f}}(1+L_{\boldsymbol{\pi}})t} \int_0^t \|\boldsymbol{\sigma}_{n-1}\left(\boldsymbol{z}_n(s)\right)\| \; ds \tag{8}$$

*and*

$$\|\widehat{\boldsymbol{x}}_n(t) - \boldsymbol{x}_n(t)\| \leq 2\beta_n e^{L_{\boldsymbol{f}}(1+L_{\boldsymbol{\pi}})t} \int_0^t \|\boldsymbol{\sigma}_{n-1}\left(\widehat{\boldsymbol{z}}_n(s)\right)\| \; ds. \tag{9}$$

*Proof.*

$$
\begin{aligned}
\|\widehat{\boldsymbol{x}}_n(t) - \boldsymbol{x}_n(t)\| &= \left\| \int_0^t \boldsymbol{f}_{n-1}(\widehat{\boldsymbol{z}}_n(s)) - \boldsymbol{f}^*(\boldsymbol{z}_n(s)) \, ds \right\| \\
&\leq \int_0^t \|\boldsymbol{f}_{n-1}(\widehat{\boldsymbol{z}}_n(s)) - \boldsymbol{f}^*(\boldsymbol{z}_n(s))\| \; ds \\
&\leq \int_0^t \|\boldsymbol{f}_{n-1}(\widehat{\boldsymbol{z}}_n(s)) - \boldsymbol{f}_{n-1}(\boldsymbol{z}_n(s))\| + \|\boldsymbol{f}_{n-1}(\boldsymbol{z}_n(s)) - \boldsymbol{f}^*(\boldsymbol{z}_n(s))\| \; ds \\
&\leq L_{\boldsymbol{f}}(1 + L_{\boldsymbol{\pi}}) \int_0^t \|\boldsymbol{x}_n(s) - \widehat{\boldsymbol{x}}_n(s)\| \; ds + 2\beta_n \int_0^t \|\boldsymbol{\sigma}_{n-1}\left(\boldsymbol{z}_n(s)\right)\| \; ds
\end{aligned}
$$

where the final inequality follows from Lemma 15 and Definition 2 (with high probability). We obtain Equation (8) by applying Grönwall's inequality (Fact 14).

Equation (9) is obtained analogously by expanding with $\boldsymbol{f}^*(\widehat{\boldsymbol{z}}_n(s))$ rather than $\boldsymbol{f}_{n-1}(\boldsymbol{z}_n(s))$. $\square$

**Lemma 5.** *For any MSS $S$, the regret of any episode $n$ is bounded with high probability by*

$$r_n(S) \leq 2\beta_n L_c(1 + L_{\boldsymbol{\pi}}) T e^{L_{\boldsymbol{f}}(1+L_{\boldsymbol{\pi}})T} \int_0^T \|\boldsymbol{\sigma}_{n-1}\left(\boldsymbol{z}_n(t)\right)\| \; dt. \tag{10}$$

*Proof.* By definition of $\boldsymbol{\pi}_n$ in OCoRL we have with high probability, $C(\boldsymbol{f}^*, \boldsymbol{\pi}^*) \geq C(\boldsymbol{f}_{n-1}, \boldsymbol{\pi}_n)$. Therefore, with high probability,

$$
\begin{aligned}
r_n(S) &= C(\boldsymbol{f}^*, \boldsymbol{\pi}_n) - C(\boldsymbol{f}^*, \boldsymbol{\pi}^*) \\
&\leq C(\boldsymbol{f}^*, \boldsymbol{\pi}_n) - C(\boldsymbol{f}_{n-1}, \boldsymbol{\pi}_n) \\
&= \int_0^T c(\boldsymbol{z}_n(t)) - c(\widehat{\boldsymbol{z}}_n(t)) \, dt.
\end{aligned}
$$

By Lemma 15, we further have that

$$r_n(S) \leq L_c(1 + L_{\boldsymbol{\pi}}) \int_0^T \|\boldsymbol{x}_n(t) - \widehat{\boldsymbol{x}}_n(t)\| \; dt.$$

By Lemma 4 (with high probability), we further have that

$$r_n(S) \leq 2\beta_n L_c(1 + L_{\boldsymbol{\pi}})e^{L_{\boldsymbol{f}}(1+L_{\boldsymbol{\pi}})T} \int_0^T \int_0^t \|\boldsymbol{\sigma}_{n-1}(\boldsymbol{z}_n(s))\| \, ds \, dt$$

$$\leq 2\beta_n L_c(1 + L_{\boldsymbol{\pi}})e^{L_{\boldsymbol{f}}(1+L_{\boldsymbol{\pi}})T} \int_0^T \int_0^T \|\boldsymbol{\sigma}_{n-1}(\boldsymbol{z}_n(s))\| \, ds \, dt$$

$$= 2\beta_n L_c(1 + L_{\boldsymbol{\pi}})Te^{L_{\boldsymbol{f}}(1+L_{\boldsymbol{\pi}})T} \int_0^T \|\boldsymbol{\sigma}_{n-1}(\boldsymbol{z}_n(t))\| \, dt.$$

$\square$

Now we are ready to prove Proposition 1.

*Proof of Proposition 1.* Let us first bound $R_N^2(S)$. By the Cauchy-Schwarz inequality,

$$R_N^2(S) \leq N \sum_{n=1}^N r_n^2(S).$$

By Lemma 5 (with high probability), we have that

$$R_N^2(S) \leq N4L_c^2(1 + L_{\boldsymbol{\pi}})^2 T^2 e^{2L_{\boldsymbol{f}}(1+L_{\boldsymbol{\pi}})T} \sum_{n=1}^N \beta_n^2 \left( \int_0^T \|\boldsymbol{\sigma}_{n-1}(\boldsymbol{z}_n(t))\| \, dt \right)^2$$

$$\leq N4\beta_N^2 L_c^2(1 + L_{\boldsymbol{\pi}})^2 T^3 e^{2L_{\boldsymbol{f}}(1+L_{\boldsymbol{\pi}})T} \sum_{n=1}^N \int_0^T \|\boldsymbol{\sigma}_{n-1}(\boldsymbol{z}_n(t))\|^2 \, dt.$$

Taking the square root, we obtain

$$R_N(S) \leq 2\beta_N L_c(1 + L_{\boldsymbol{\pi}})T^{\frac{3}{2}}e^{L_{\boldsymbol{f}}(1+L_{\boldsymbol{\pi}})T} \sqrt{N \sum_{n=1}^N \int_0^T \|\boldsymbol{\sigma}_{n-1}(\boldsymbol{z}_n(t))\|^2 \, dt}.$$

The result follows by noting that

$$\sum_{n=1}^N \int_0^T \|\boldsymbol{\sigma}_{n-1}(\boldsymbol{z}_n(t))\|^2 \, dt \leq \mathcal{I}_N(\boldsymbol{f}^*, S).$$

$\square$

## A.2   Bounding Model Complexities

In this section, we derive the model complexity bounds for the presented MSSs. Our bounds are based on the approximation of integrals using an upper Darboux sum.

**Fact 6** (Upper Darboux approximation). *Given any $a \leq b$, a function $f : \mathbb{R} \to \mathbb{R}$, and a partition $P = (a, \ldots, b)$ of $[a, b]$ into $k$ sub-intervals,*

$$\int_a^b f(x) \, dx \leq \sum_{i=1}^k \Delta_i \max_{x \in P[i]} f(x) \tag{11}$$

*where $P[i]$ denotes the $i$-th sub-interval and $\Delta_i$ denotes the length of $P[i]$.*

We are now ready to bound the model complexities.

**Lemma 7** (Oracle model complexity). *For any $N \geq 1$,*

$$\mathcal{I}_N(\boldsymbol{f}^*, S^{ORA}) \leq \max_{\substack{\boldsymbol{\pi}_1,\ldots,\boldsymbol{\pi}_N \\ \boldsymbol{\pi}_n \in \Pi}} T \sum_{n=1}^N \|\boldsymbol{\sigma}_{n-1}(\boldsymbol{z}_n(t_{n,1}))\|^2. \tag{12}$$

*Proof.*

$$\mathcal{I}_N(\boldsymbol{f}^*, S^{\text{ORA}}) = \max_{\substack{\boldsymbol{\pi}_1,\ldots,\boldsymbol{\pi}_N \\ \boldsymbol{\pi}_n \in \Pi}} \sum_{n=1}^N \int_0^T \|\boldsymbol{\sigma}_{n-1}(\boldsymbol{z}_n(t))\|^2 \ dt.$$

For each episode $n$, use an upper Darboux approximation (Fact 6) with $k = 1$ to obtain

$$\leq \max_{\substack{\boldsymbol{\pi}_1,\ldots,\boldsymbol{\pi}_N \\ \boldsymbol{\pi}_n \in \Pi}} T \sum_{n=1}^N \max_{0 \leq t \leq T} \|\boldsymbol{\sigma}_{n-1}(\boldsymbol{z}_n(t))\|^2$$

$$= \max_{\substack{\boldsymbol{\pi}_1,\ldots,\boldsymbol{\pi}_N \\ \boldsymbol{\pi}_n \in \Pi}} T \sum_{n=1}^N \|\boldsymbol{\sigma}_{n-1}(\boldsymbol{z}_n(t_{n,1}))\|^2 .$$

$\square$

**Lemma 8** (Equidistant model complexity). *For any $N \geq 1$,*

$$\mathcal{I}_N(\boldsymbol{f}^*, S^{EQI}) \leq \max_{\substack{\boldsymbol{\pi}_1,\ldots,\boldsymbol{\pi}_N \\ \boldsymbol{\pi}_n \in \Pi}} \sum_{n=1}^N \frac{T}{m_n} \sum_{i=1}^{m_n} \left( \|\boldsymbol{\sigma}_{n-1}(\boldsymbol{z}_n(t_{n,i}))\|^2 + \frac{TL_{\sigma^2}}{m_n} \right). \tag{13}$$

*Proof.*

$$\mathcal{I}_N(\boldsymbol{f}^*, S^{\text{EQI}}) = \max_{\substack{\boldsymbol{\pi}_1,\ldots,\boldsymbol{\pi}_N \\ \boldsymbol{\pi}_n \in \Pi}} \sum_{n=1}^N \int_0^T \|\boldsymbol{\sigma}_{n-1}(\boldsymbol{z}_n(t))\|^2 \ dt.$$

For each episode $n$, let $P_n \stackrel{\text{def}}{=} (0, \Delta_n, 2\Delta_n, \ldots, T)$ with $\Delta_n \stackrel{\text{def}}{=} \frac{T}{m_n}$ be a partition of $[0, T]$ into $m_n$ sub-intervals, each of length $\Delta_n$. Using the upper Darboux approximations (Fact 6) of $\|\boldsymbol{\sigma}_{n-1}(\cdot)\|^2$ with respect to $P_n$, we have

$$\leq \max_{\substack{\boldsymbol{\pi}_1,\ldots,\boldsymbol{\pi}_N \\ \boldsymbol{\pi}_n \in \Pi}} \sum_{n=1}^N \Delta_n \sum_{i=1}^{m_n} \max_{t \in [(i-1)\Delta_n, i\Delta_n]} \|\boldsymbol{\sigma}_{n-1}(\boldsymbol{z}_n(t))\|^2 .$$

Observe that $t_{n,i} \in [(i-1)\Delta_n, i\Delta_n]$, and hence, using that $\|\boldsymbol{\sigma}_{n-1}(\cdot)\|^2$ is $L_{\sigma^2}$-Lipschitz continuous,

$$\leq \max_{\substack{\boldsymbol{\pi}_1,\ldots,\boldsymbol{\pi}_N \\ \boldsymbol{\pi}_n \in \Pi}} \sum_{n=1}^N \Delta_n \sum_{i=1}^{m_n} \left( \|\boldsymbol{\sigma}_{n-1}(\boldsymbol{z}_n(t_{n,i}))\|^2 + L_{\sigma^2}\Delta_n \right).$$

$\square$

In the remainder of this section, we study the model complexity of the adaptive MSS. In each episode $n$, we partition $[0, T]$ into equally sized sub-intervals (called "buckets") of length $\Delta_n$, where $\Delta_n$ is the solution to

$$\Delta_n = \frac{1}{2\Gamma_n(\Delta_n)} \tag{14}$$

where $\Gamma_n(\Delta_n) \stackrel{\text{def}}{=} 2\beta_n L_{\boldsymbol{\sigma}}(1 + L_{\boldsymbol{\pi}})e^{L_{\boldsymbol{f}}(1+L_{\boldsymbol{\pi}})\Delta_n}$. As $\Gamma_n$ is a monotonically increasing function of $\Delta_n$ it is clear that a suitable $\Delta_n$ exists. Moreover, it follows from $\Delta_n \leq T$ that $\Gamma_n \in \mathcal{O}(\beta_n)$, and hence that $m_n = T/\Delta_n \in \mathcal{O}(\beta_n)$. Without loss of generality, we assume $m_n \in \mathbb{N}$.

The definition of the adaptive MSS can be interpreted as follows: In every bucket, we take two measurements. One measurement is saved and used later for updating the statistical model. The other measurement is taken at the beginning of every bucket, and used to inform OCoRL of the current state position. For the purposes of our theoretical analysis, these initial measurements are not saved and not used for the construction of the statistical model.

**Lemma 9.** *For any episode $n$, bucket $i \in \{1, \ldots, m_n\}$, and $t \in [(i-1)\Delta_n, i\Delta_n]$ where the real and hallucinated trajectories are "synced" initially, i.e., $\widehat{\boldsymbol{z}}_n((i-1)\Delta_n) = \boldsymbol{z}_n((i-1)\Delta_n)$, we have with high probability that*

$$\|\boldsymbol{\sigma}_{n-1}(\widehat{\boldsymbol{z}}_n(t)) - \boldsymbol{\sigma}_{n-1}(\boldsymbol{z}_n(t))\| \le \Gamma_n \int_{(i-1)\Delta_n}^{t} \|\boldsymbol{\sigma}_{n-1}(\boldsymbol{z}_n(s))\| \, ds \qquad and \qquad (15)$$

$$\|\boldsymbol{\sigma}_{n-1}(\widehat{\boldsymbol{z}}_n(t)) - \boldsymbol{\sigma}_{n-1}(\boldsymbol{z}_n(t))\| \le \Gamma_n \int_{(i-1)\Delta_n}^{t} \|\boldsymbol{\sigma}_{n-1}(\widehat{\boldsymbol{z}}_n(s))\| \, ds \qquad (16)$$

*Proof.* Using that $\boldsymbol{\sigma}_{n-1}$ is $L_{\boldsymbol{\sigma}}$-Lipschitz continuous and applying Lemma 15,

$$\|\boldsymbol{\sigma}_{n-1}(\widehat{\boldsymbol{z}}_n(t)) - \boldsymbol{\sigma}_{n-1}(\boldsymbol{z}_n(t))\| \le L_{\boldsymbol{\sigma}}(1 + L_{\boldsymbol{\pi}}) \|\widehat{\boldsymbol{x}}_n(t) - \boldsymbol{x}_n(t)\|.$$

The result then follows from Lemma 4, where the lower integral bounds can be tightened to $(i-1)\Delta_n$ as the real and hallucinated trajectories are "synced" in the beginning of the bucket. $\square$

**Lemma 10.** *For any episode $n$ and $i \in \{1, \ldots, m_n\}$, if $t_{n,i}$ is selected according to the adaptive MSS then with high probability,*

$$\max_{t \in [(i-1)\Delta_n, i\Delta_n]} \|\boldsymbol{\sigma}_{n-1}(\boldsymbol{z}_n(t))\|^2 \le 9 \|\boldsymbol{\sigma}_{n-1}(\boldsymbol{z}_n(t_{n,i}))\|^2. \qquad (17)$$

*Proof.* By applying Lemma 9 we have (with high probability) for every $t \in [(i-1)\Delta_n, \Delta_n]$,

$$\|\boldsymbol{\sigma}_{n-1}(\boldsymbol{z}_n(t))\| \le \|\boldsymbol{\sigma}_{n-1}(\widehat{\boldsymbol{z}}_n(t))\| + \Gamma_n \int_{(i-1)\Delta_n}^{t} \|\boldsymbol{\sigma}_{n-1}(\widehat{\boldsymbol{z}}_n(s))\| \, ds.$$

Note that $\|\boldsymbol{\sigma}_{n-1}(\widehat{\boldsymbol{z}}_n(t))\| \le \|\boldsymbol{\sigma}_{n-1}(\widehat{\boldsymbol{z}}_n(t_{n,i}))\|$ by the definition of $t_{n,i}$, and hence,

$$\|\boldsymbol{\sigma}_{n-1}(\boldsymbol{z}_n(t))\| \le \|\boldsymbol{\sigma}_{n-1}(\widehat{\boldsymbol{z}}_n(t_{n,i}))\| + \Gamma_n \Delta_n \|\boldsymbol{\sigma}_{n-1}(\widehat{\boldsymbol{z}}_n(t_{n,i}))\|$$
$$= \left(1 + \frac{1}{2}\right) \|\boldsymbol{\sigma}_{n-1}(\widehat{\boldsymbol{z}}_n(t_{n,i}))\|. \qquad (18)$$

Similarly, by applying Lemma 9, we have (with high probability) that

$$\|\boldsymbol{\sigma}_{n-1}(\boldsymbol{z}_n(t_{n,i}))\| \ge \|\boldsymbol{\sigma}_{n-1}(\widehat{\boldsymbol{z}}_n(t_{n,i}))\| - \Gamma_n \int_{(i-1)\Delta_n}^{t_{n,i}} \|\boldsymbol{\sigma}_{n-1}(\widehat{\boldsymbol{z}}_n(s))\| \, ds$$
$$\ge \|\boldsymbol{\sigma}_{n-1}(\widehat{\boldsymbol{z}}_n(t_{n,i}))\| - \Gamma_n \Delta_n \|\boldsymbol{\sigma}_{n-1}(\widehat{\boldsymbol{z}}_n(t_{n,i}))\|$$
$$= \left(1 - \frac{1}{2}\right) \|\boldsymbol{\sigma}_{n-1}(\widehat{\boldsymbol{z}}_n(t_{n,i}))\|. \qquad (19)$$

Combining Equations (18) and (19), we obtain

$$\|\boldsymbol{\sigma}_{n-1}(\boldsymbol{z}_n(t))\|^2 \le \left(\frac{1 + \frac{1}{2}}{1 - \frac{1}{2}}\right)^2 \|\boldsymbol{\sigma}_{n-1}(\boldsymbol{z}_n(t_{n,i}))\|^2 = 9 \|\boldsymbol{\sigma}_{n-1}(\boldsymbol{z}_n(t_{n,i}))\|^2.$$

$\square$

**Corollary 11** (Adaptive model complexity). *For any $N \ge 1$,*

$$\mathcal{I}_N(\boldsymbol{f}^*, S^{ADP}) \le \max_{\substack{\boldsymbol{\pi}_1, \ldots, \boldsymbol{\pi}_N \\ \boldsymbol{\pi}_n \in \Pi}} 9 \sum_{n=1}^{N} \frac{T}{m_n} \sum_{i=1}^{m_n} \|\boldsymbol{\sigma}_{n-1}(\boldsymbol{z}_n(t_{n,i}))\|^2. \qquad (20)$$

*Proof.* The result follows analogously to the proof of Lemma 8, using Lemma 10 in the last step. $\square$

## A.3 Bounding Measurement Uncertainty of GPs

Our goal in this section is to prove Theorem 3.

The informativeness of a set of sampling points $A \subset \mathcal{Z}$ about a function $f \sim \mathcal{GP}(0, k)$ with the noisy observations $\boldsymbol{y}_A = \boldsymbol{f}_A + \boldsymbol{\epsilon}_A$ can be measured by the *information gain* (c.f., Cover (1999); Srinivas et al. (2009)),

$$\mathrm{I}(\boldsymbol{y}_A; f) \stackrel{\text{def}}{=} \mathrm{I}(\boldsymbol{y}_A; \boldsymbol{f}_A) = \mathrm{H}(\boldsymbol{y}_A) - \mathrm{H}(\boldsymbol{y}_A \mid \boldsymbol{f}_A), \tag{21}$$

which quantifies the reduction in entropy of $\boldsymbol{f}_A$ when observing $\boldsymbol{y}_A$. Here, we write $\boldsymbol{f}_A = [f(\boldsymbol{x})]_{\boldsymbol{x} \in A}$ and $\boldsymbol{\epsilon}_A \sim \mathcal{N}\left(\boldsymbol{0}, \sigma^2 \boldsymbol{I}\right)$. For a Gaussian, $\mathrm{H}(\mathcal{N}\left(\boldsymbol{\mu}, \boldsymbol{\Sigma}\right)) = \frac{1}{2} \log \det(2\pi e \boldsymbol{\Sigma})$, so that when $f$ is a Gaussian process, $\mathrm{I}(\boldsymbol{y}_A; f) = \frac{1}{2} \log \det(\boldsymbol{I} + \sigma^{-2} \boldsymbol{K}_{AA})$ where $\boldsymbol{K}_{AA} = [k(\boldsymbol{x}, \boldsymbol{x}')]_{\boldsymbol{x}, \boldsymbol{x}' \in A}$. We denote the maximal information gain by observing $n$ points by

$$\gamma_n \stackrel{\text{def}}{=} \max_{\substack{A \subset \mathcal{Z} \\ |A|=n}} \mathrm{I}(\boldsymbol{y}_A; f). \tag{22}$$

Clearly, $\gamma_n$ depends on the kernel $k$. In Table 2, we state the magnitude of $\gamma_n$ for different kernels. We take the magnitudes from Theorem 5 of Srinivas et al. (2009) and Remark 2 of Vakili et al. (2021).

Table 2: Here we present different magnitudes of $\gamma_n$. The magnitudes hold under the assumption that $\mathcal{Z}$ is compact. Here, $B_\nu$ is the modified Bessel function.

| Kernel | $k(x, x')$ | $\gamma_n$ |
|---|---|---|
| Linear | $x^\top x'$ | $\mathcal{O}\left(d \log(n)\right)$ |
| RBF | $e^{-\frac{\|x-x'\|^2}{2l^2}}$ | $\mathcal{O}\left(\log^{d+1}(n)\right)$ |
| Matérn | $\frac{1}{\Gamma(\nu)2^{\nu-1}} \left(\frac{\sqrt{2\nu}\|x-x'\|}{l}\right)^\nu B_\nu \left(\frac{\sqrt{2\nu}\|x-x'\|}{l}\right)$ | $\mathcal{O}\left(n^{\frac{d}{2\nu+d}} \log^{\frac{2\nu}{2\nu+d}}(n)\right)$ |

The maximum information gain can be interpreted analogously in the setting where $f \in \mathcal{H}_{k,B}$, see Srinivas et al. (2009) and section 5.2 of Kirschner and Krause (2018). The following result relates mutual information and epistemic uncertainty.

**Lemma 12** (Lemma 5.3 from Srinivas et al. (2009)). *For any $A \subset \mathcal{Z}$, if $f \sim \mathcal{GP}(0, k)$ or $f \in \mathcal{H}_{k,B}$, and if the observation noise is i.i.d. $\mathcal{N}(0; \sigma^2)$ then*

$$\mathrm{I}(\boldsymbol{y}_A; f) = \frac{1}{2} \sum_{i=1}^{n} \log\left(1 + \frac{\mathrm{Var}[f(\boldsymbol{x}_i) \mid \boldsymbol{y}(\boldsymbol{x}_{1:i-1})]}{\sigma^2}\right) \tag{23}$$

*where $\{\boldsymbol{x}_1, \ldots, \boldsymbol{x}_n\} \stackrel{\text{def}}{=} A$.*

In our setting, denote by $A_N \stackrel{\text{def}}{=} (S_1, \ldots, S_N)$ the set of times of observations $\mathcal{D}_{1:N}$ until episode $N$. We define for every episode $n$,

$$i_n^* \stackrel{\text{def}}{=} \underset{i \in \{1, \ldots, m_n\}}{\mathrm{argmax}} \|\boldsymbol{\sigma}_{n-1}(\boldsymbol{z}_n(t_{n,i}))\|_2^2. \tag{24}$$

We write $t_n^* \stackrel{\text{def}}{=} t_{n,i_n^*}$ and $\tilde{A}_N \stackrel{\text{def}}{=} (\{t_1^*\}, \ldots, \{t_N^*\})$. Note that $\tilde{A}_N[n] \subseteq A_N[n]$ ($\forall n$) and $\tilde{A}_N$ comprises exactly $N$ observations.

**Lemma 13.** *If $f_j \sim \mathcal{GP}(0, k)$ for all $j \in \{1, \ldots, d_x\}$ or if $\boldsymbol{f} \in \mathcal{H}_{k,B}^{d_x}$, and if the observation noise is i.i.d. $\mathcal{N}(\boldsymbol{0}; \sigma^2 \boldsymbol{I})$ then*

$$\sum_{n=1}^{N} \frac{T}{m_n} \sum_{i=1}^{m_n} \|\boldsymbol{\sigma}_{n-1}(\boldsymbol{z}_n(t_{n,i}))\|_2^2 \leq 2\bar{\sigma} T d_x \gamma_N \tag{25}$$

*where $\bar{\sigma} = \frac{\sigma_{\max}^2}{\log(1+\sigma^{-2}\sigma_{\max}^2)}$, $\sigma_{\max}^2 = \max_{\boldsymbol{z} \in \mathcal{Z}, j \in \{1, \ldots, d_x\}} \sigma_{0,j}^2(\boldsymbol{z})$, and $\gamma_N$ is with respect to kernel $k$.*

*Proof.* We have

$$\sum_{n=1}^{N} \frac{T}{m_n} \sum_{i=1}^{m_n} \|\boldsymbol{\sigma}_{n-1}(\boldsymbol{z}_n(t_{n,i}))\|_2^2 \le T \sum_{n=1}^{N} \|\boldsymbol{\sigma}_{n-1}(\boldsymbol{z}_n(t_n^*))\|_2^2.$$

This inequality is tight when $m_n = 1$ ($\forall n$). When more than one measurement is obtained per episode then the regret may be smaller. Expanding the squared Euclidean norm, yields

$$= T \sum_{n=1}^{N} \sum_{j=1}^{d_x} \sigma_{n-1,j}^2(\boldsymbol{z}_n(t_n^*)).$$

Write $\tilde{\sigma}_{n,j}^2(\boldsymbol{z}) \stackrel{\text{def}}{=} \mathrm{Var}[f_j(\boldsymbol{z}) \mid \dot{\boldsymbol{y}}_{\tilde{A}_N}]$ whereas $\sigma_{n,j}^2(\boldsymbol{z}) = \mathrm{Var}[f_j(\boldsymbol{z}) \mid \dot{\boldsymbol{y}}_{A_N}]$. The variance is monotonically decreasing as one conditions on more observations, $\tilde{\sigma}_{n,j}^2(\boldsymbol{z}) \ge \sigma_{n,j}^2(\boldsymbol{z})$ ($\forall n, j, \boldsymbol{z}$), and hence,

$$\le T \sum_{n=1}^{N} \sum_{j=1}^{d_x} \tilde{\sigma}_{n-1,j}^2(\boldsymbol{z}_n(t_n^*)).$$

Again, this bound is tight when only one measurement is obtained per episode, but may be loose otherwise. Lemma 15 of Curi et al. (2020) shows that $\tilde{\sigma}_{n,j}^2(\boldsymbol{z}) \le \bar{\sigma} \log(1 + \sigma^{-2}\tilde{\sigma}_{n,j}^2(\boldsymbol{z}))$ for any $n, j, \boldsymbol{z}$. Applying this inequality, we obtain

$$\le \bar{\sigma} T \sum_{j=1}^{d_x} \sum_{n=1}^{N} \log\left(1 + \frac{\tilde{\sigma}_{n-1,j}^2(\boldsymbol{z}_n(t_n^*))}{\sigma^2}\right).$$

By Lemma 12,

$$= 2\bar{\sigma} T \sum_{j=1}^{d_x} \mathrm{I}\big(\dot{\boldsymbol{y}}_{\tilde{A}_N}; f_j\big)$$

$$\le 2\bar{\sigma} T d_x \gamma_N.$$

$\square$

*Proof of Theorem 3.* We derive the regret bound for the adaptive MSS. The regret bounds for the other MSSs follow analogously.

From Corollary 11, recall the model complexity bound

$$\mathcal{I}_N(\boldsymbol{f}^*, S^{\mathrm{ADP}}) \le \max_{\substack{\boldsymbol{\pi}_1,\dots,\boldsymbol{\pi}_N \\ \boldsymbol{\pi}_n \in \Pi}} 9 \sum_{n=1}^{N} \frac{T}{m_n} \sum_{i=1}^{m_n} \|\boldsymbol{\sigma}_{n-1}(\boldsymbol{z}_n(t_{n,i}))\|^2.$$

By Lemma 13, we can further bound this by

$$\le 18\bar{\sigma} T d_x \gamma_N.$$

Combining this bound with Proposition 1, with high probability the regret is bounded by

$$R_N(S^{\mathrm{ADP}}) \le \mathcal{O}\Big(\beta_N T^2 e^{L_f(1+L_{\boldsymbol{\pi}})T} \sqrt{N \gamma_N}\Big).$$

$\square$

### A.4 Useful Facts and Inequalities

**Fact 14** (Grönwall's inequality, theorem 1.1 in chapter 3 of Hartman (2002))**.** *Let $g(t)$ be a non-negative continuous function on the interval $[a, b] \subset \mathbb{R}$, and let $C, K$ be a pair of non-negative constants. If the function $g$ satisfies*

$$g(t) \le C + K \int_a^t g(s) \, ds, \quad \forall t \in [a, b], \tag{26}$$

*then we have*

$$g(t) \le C e^{K(t-a)}, \quad \forall t \in [a, b]. \tag{27}$$

**Lemma 15.** *For any two metric spaces $(\mathcal{X}, d_\mathcal{X})$ and $(\mathcal{Y}, d_\mathcal{Y})$, and any two Lipschitz continuous functions $f : \mathcal{X} \times \mathcal{X} \to \mathcal{Y}$ and $g : \mathcal{X} \to \mathcal{X}$ with Lipschitz constants $L_f$ and $L_g$ respectively, we have that $f(\cdot, g(\cdot))$ is L-Lipschitz continuous with respect to $d_\mathcal{Y}$ with $L \stackrel{def}{=} L_f(1 + L_g)$.*

*Proof.* Fix any $\boldsymbol{x}, \boldsymbol{x}' \in \mathcal{X}$. By the triangle inequality,

$$
\begin{aligned}
d_\mathcal{Y}(f(\boldsymbol{x}, g(\boldsymbol{x})), f(\boldsymbol{x}', g(\boldsymbol{x}'))) &\leq d_\mathcal{Y}(f(\boldsymbol{x}', g(\boldsymbol{x})), f(\boldsymbol{x}, g(\boldsymbol{x}))) + d_\mathcal{Y}(f(\boldsymbol{x}', g(\boldsymbol{x})), f(\boldsymbol{x}', g(\boldsymbol{x}'))) \\
&\leq L_f d_\mathcal{X}(\boldsymbol{x}, \boldsymbol{x}') + L_f d_\mathcal{X}(g(\boldsymbol{x}), g(\boldsymbol{x}')) \\
&\leq L_f d_\mathcal{X}(\boldsymbol{x}, \boldsymbol{x}') + L_f L_g d_\mathcal{X}(\boldsymbol{x}, \boldsymbol{x}') \\
&= L_f(1 + L_g) d_\mathcal{X}(\boldsymbol{x}, \boldsymbol{x}').
\end{aligned}
$$

$\square$

# B    Solving Optimistic Optimal Control Problem

The optimal control problem solved by OCoRL is constrained to $L_{\boldsymbol{f}}$-Lipschitz continuous dynamics. When the dynamics are modeled with a GP using a linear kernel, the control problem can easily be solved directly in weight-space. Beyond this special case, it is commonly assumed that there exists an oracle which solves this optimization problem exactly (see, e.g., assumption 1 of Kakade et al. (2020)).

In this work, we do not explicitly address the problem of approximating the optimal control problem reasonably well. Kakade et al. (2020) give a brief overview of gradient- and sampling-based methods which may be useful (Jacobson and Mayne, 1970; Todorov and Li, 2005; Mordatch et al., 2012; Williams et al., 2017; Wagener et al., 2019). They alternatively suggest to use Thompson sampling (Thompson, 1933; Osband and Van Roy, 2014), that is, to sample $\boldsymbol{f}_{n-1}$ from $\mathcal{M}_{n-1}$ (which is straightforward when the dynamics are modeled by a GP, see Williams and Rasmussen (2006)) and then to compute and execute the optimal policy $\boldsymbol{\pi}_n = \operatorname{argmin}_{\boldsymbol{\pi} \in \Pi} C(\boldsymbol{\pi}, \boldsymbol{f}_{n-1})$ using a planning oracle. Kakade et al. (2020) conjecture that corresponding regret bounds can be derived using standard techniques for analyzing Bayesian regret of Thompson sampling (Russo and Van Roy, 2014, 2016).

## B.1    Application to Arbitrary Discretizations

In this section, we briefly show that our guarantees carry over to discretized dynamics and costs for *arbitrary* discretizations. Importantly, in the discrete-time setting the lemma mirroring Lemma 4 does not require the assumption that the models $\boldsymbol{f}_n$ are $L_{\boldsymbol{f}}$-Lipschitz continuous, hence, simplifying the optimistic optimal control problem solved by OCoRL. In the discrete-time setting, it is sufficient to assume that $\boldsymbol{f}^*$ is $L_{\boldsymbol{f}}$-Lipschitz continuous.

Consider the dynamical system

$$
\boldsymbol{x}_{k+1} = \boldsymbol{h}^*(\boldsymbol{z}_k, \tau_k), \quad \text{where} \quad \boldsymbol{z}_k \stackrel{def}{=} (\boldsymbol{x}_k, \boldsymbol{\pi}(\boldsymbol{x}_k))
$$

with initial state $\boldsymbol{x}_0 \in \mathcal{X}$ where $\tau_k \in \mathbb{R}_{>0}$ measures the "time" between control points $\boldsymbol{x}_k$ and $\boldsymbol{x}_{k+1}$. We make the assumption that $\boldsymbol{h}^*$ is $L_{\boldsymbol{h}}$-Lipschitz continuous in all arguments. The (discrete-time) control problem is

$$
\boldsymbol{\pi}^* \stackrel{def}{=} \operatorname*{argmin}_{\boldsymbol{\pi} \in \Pi} C(\boldsymbol{\pi}, \boldsymbol{h}^*) = \operatorname*{argmin}_{\boldsymbol{\pi} \in \Pi} \sum_{k=1}^{T} c(\boldsymbol{z}_k). \tag{28}
$$

In many applications, the cost to be minimized is defined at discrete control points to begin with. In this case, the continuous-time control of Equation (1) can be approximated arbitrarily well with Equation (28) by choosing a sufficiently dense discretization. Commonly, the discretization is taken to be equidistant, i.e., $\tau_k \equiv \tau \; (\forall k)$, but we remark that our results hold more generally for any discretization.

### B.1.1 Bounding Regret with the Model Complexity

In the following, we denote by $\boldsymbol{x}_{n,k}$ the $k$-th state visited during episode $n$.

**Lemma 16.** *Assuming $\boldsymbol{x}_{n,0} = \widehat{\boldsymbol{x}}_{n,0}$, the distance between the true and the hallucinated trajectory at any iteration $k \geq 0$ is bounded with high probability by*

$$\|\widehat{\boldsymbol{x}}_{n,k} - \boldsymbol{x}_{n,k}\| \leq 2\beta_n(1 + L'_{\boldsymbol{h}} + 2\beta_n L'_{\boldsymbol{\sigma}})^{T-1} \sum_{l=0}^{k-1} \|\boldsymbol{\sigma}_{n-1}(\boldsymbol{z}_{n,l})\| \tag{29}$$

*where we write $L'_{\boldsymbol{h}} \stackrel{def}{=} L_{\boldsymbol{h}}(1 + L_{\boldsymbol{\pi}})$ and $L'_{\boldsymbol{\sigma}} \stackrel{def}{=} L_{\boldsymbol{\sigma}}(1 + L_{\boldsymbol{\pi}})$*

*Proof (based on Lemma 4 of Curi et al. (2020)).* We begin by showing by induction that for any $k \geq 0$,

$$\|\widehat{\boldsymbol{x}}_{n,k} - \boldsymbol{x}_{n,k}\| \leq 2\beta_n \sum_{l=0}^{k-1} (L'_{\boldsymbol{h}} + 2\beta_n L'_{\boldsymbol{\sigma}})^{k-1-l} \|\boldsymbol{\sigma}_{n-1}(\boldsymbol{z}_{n,l})\| . \tag{30}$$

The base case is implied trivially. For the induction step, assume that Equation (30) holds at iteration $k$. We have

$$\begin{aligned}
\|\widehat{\boldsymbol{x}}_{n,k+1} - \boldsymbol{x}_{n,k+1}\| &= \|\boldsymbol{h}_{n-1}(\widehat{\boldsymbol{z}}_{n,k}, \tau_k) - \boldsymbol{h}^*(\boldsymbol{z}_{n,k}, \tau_k)\| \\
&\leq \|\boldsymbol{h}_{n-1}(\widehat{\boldsymbol{z}}_{n,k}, \tau_k) - \boldsymbol{h}^*(\boldsymbol{z}_{n,k}, \tau_k)\| \\
&\leq \|\boldsymbol{h}_{n-1}(\widehat{\boldsymbol{z}}_{n,k}, \tau_k) - \boldsymbol{h}^*(\widehat{\boldsymbol{z}}_{n,k}, \tau_k)\| + \|\boldsymbol{h}^*(\widehat{\boldsymbol{z}}_{n,k}, \tau_k) - \boldsymbol{h}^*(\boldsymbol{z}_{n,k}, \tau_k)\| \\
&\leq 2\beta_n \|\boldsymbol{\sigma}_{n-1}(\widehat{\boldsymbol{z}}_{n,k})\| + L_{\boldsymbol{h}}(1 + L_{\boldsymbol{\pi}}) \|\boldsymbol{x}_{n,k} - \widehat{\boldsymbol{x}}_{n,k}\|
\end{aligned}$$

where the final inequality follows from Definition 2 (with high probability) and Lemma 15.

$$\begin{aligned}
&= 2\beta_n \|\boldsymbol{\sigma}_{n-1}(\boldsymbol{z}_{n,k}) + \boldsymbol{\sigma}_{n-1}(\widehat{\boldsymbol{z}}_{n,k}) - \boldsymbol{\sigma}_{n-1}(\boldsymbol{z}_{n,k})\| \\
&\qquad\qquad + L_{\boldsymbol{h}}(1 + L_{\boldsymbol{\pi}}) \|\boldsymbol{x}_{n,k} - \widehat{\boldsymbol{x}}_{n,k}\|
\end{aligned}$$

Using that $\boldsymbol{\sigma}_{n-1}$ is $L_{\boldsymbol{\sigma}}$-Lipschitz continuous and applying Lemma 15,

$$\begin{aligned}
&\leq 2\beta_n(\|\boldsymbol{\sigma}_{n-1}(\boldsymbol{z}_{n,k})\| + L_{\boldsymbol{\sigma}}(1 + L_{\boldsymbol{\pi}}) \|\widehat{\boldsymbol{x}}_{n,k} - \boldsymbol{x}_{n,k}\|) \\
&\qquad\qquad + L_{\boldsymbol{h}}(1 + L_{\boldsymbol{\pi}}) \|\boldsymbol{x}_{n,k} - \widehat{\boldsymbol{x}}_{n,k}\| \\
&= 2\beta_n \|\boldsymbol{\sigma}_{n-1}(\boldsymbol{z}_{n,k})\| + (L_{\boldsymbol{h}}(1 + L_{\boldsymbol{\pi}}) + 2\beta_n L_{\boldsymbol{\sigma}}(1 + L_{\boldsymbol{\pi}})) \|\boldsymbol{x}_{n,k} - \widehat{\boldsymbol{x}}_{n,k}\|
\end{aligned}$$

By the induction hypothesis,

$$\leq 2\beta_n \sum_{l=0}^{k} (L'_{\boldsymbol{h}} + 2\beta_n L'_{\boldsymbol{\sigma}})^{k-l} \|\boldsymbol{\sigma}_{n-1}(\boldsymbol{z}_{n,l})\| .$$

Thus, Equation (30) holds. Since $k \leq T$, we have

$$\begin{aligned}
\|\widehat{\boldsymbol{x}}_{n,k} - \boldsymbol{x}_{n,k}\| &\leq 2\beta_n \sum_{l=0}^{k-1} (L'_{\boldsymbol{h}} + 2\beta_n L'_{\boldsymbol{\sigma}})^{k-1-l} \|\boldsymbol{\sigma}_{n-1}(\boldsymbol{z}_{n,l})\| \\
&\leq 2\beta_n \sum_{l=0}^{k-1} (1 + L'_{\boldsymbol{h}} + 2\beta_n L'_{\boldsymbol{\sigma}})^{k-1-l} \|\boldsymbol{\sigma}_{n-1}(\boldsymbol{z}_{n,l})\| \\
&\leq 2\beta_n(1 + L'_{\boldsymbol{h}} + 2\beta_n L'_{\boldsymbol{\sigma}})^{T-1} \sum_{l=0}^{k-1} \|\boldsymbol{\sigma}_{n-1}(\boldsymbol{z}_{n,l})\| .
\end{aligned}$$

$\square$

**Lemma 17.** *For any MSS S, the regret of any episode $n$ is bounded with high probability by*

$$r_n(S) \leq 2\beta_n T(1 + L'_{\boldsymbol{h}} + 2\beta_n L'_{\boldsymbol{\sigma}})^{T-1} L'_c \sum_{k=1}^{T} \|\boldsymbol{\sigma}_{n-1}(\boldsymbol{z}_{n,k})\| \tag{31}$$

*where $L'_c \stackrel{def}{=} L_c(1 + L_{\boldsymbol{\pi}})$.*

*Proof.* By definition of $\boldsymbol{\pi}_n$ in OCoRL we have with high probability, $C(\boldsymbol{f}^*, \boldsymbol{\pi}^*) \geq C(\boldsymbol{f}_{n-1}, \boldsymbol{\pi}_n)$. Therefore, with high probability,

$$
\begin{aligned}
r_n(S) &= C(\boldsymbol{f}^*, \boldsymbol{\pi}_n) - C(\boldsymbol{f}^*, \boldsymbol{\pi}^*) \\
&\leq C(\boldsymbol{f}^*, \boldsymbol{\pi}_n) - C(\boldsymbol{f}_{n-1}, \boldsymbol{\pi}_n) \\
&= \sum_{k=0}^{T} c(\boldsymbol{z}_{n,k}) - c(\widehat{\boldsymbol{z}}_{n,k}).
\end{aligned}
$$

By Lemma 15, we further have that

$$
r_n(S) \leq L_c' \sum_{k=1}^{T} \|\boldsymbol{x}_{n,k} - \widehat{\boldsymbol{x}}_{n,k}\|.
$$

By Lemma 16 (with high probability), we further have that

$$
\begin{aligned}
r_n(S) &\leq 2\beta_n(1 + L_{\boldsymbol{h}}' + 2\beta_n L_{\boldsymbol{\sigma}}')^{T-1} L_c' \sum_{k=1}^{T} \sum_{l=0}^{k-1} \|\boldsymbol{\sigma}_{n-1}(\boldsymbol{z}_{n,l})\| \\
&= 2\beta_n T (1 + L_{\boldsymbol{h}}' + 2\beta_n L_{\boldsymbol{\sigma}}')^{T-1} L_c' \sum_{k=1}^{T} \|\boldsymbol{\sigma}_{n-1}(\boldsymbol{z}_{n,k})\|.
\end{aligned}
$$

$\square$

The above results allow us to prove a general regret bound which is analogous to Proposition 1.

**Lemma 18.** *Let $S$ be any MSS. If we run* OCoRL *with arbitrary discretization $\boldsymbol{h}^*$, we have with high probability,*

$$
R_N(S) \leq 2\beta_N T^{\frac{3}{2}}(1 + L_{\boldsymbol{h}}' + 2\beta_N L_{\boldsymbol{\sigma}}')^{T-1} L_c' \sqrt{N \tilde{\mathcal{I}}_N(\boldsymbol{h}^*, S)} \tag{32}
$$

*where we define the discrete-time model complexity,*

$$
\tilde{\mathcal{I}}_N(\boldsymbol{h}^*, S) \stackrel{def}{=} \max_{\substack{\boldsymbol{\pi}_1, \dots, \boldsymbol{\pi}_N \\ \boldsymbol{\pi}_n \in \Pi}} \sum_{n=1}^{N} \sum_{k=1}^{T} \|\boldsymbol{\sigma}_{n-1}(\boldsymbol{z}_{n,k})\|^2. \tag{33}
$$

*Proof.* Let us first bound $R_N^2(S)$. By the Cauchy-Schwarz inequality,

$$
R_N^2(S) \leq N \sum_{n=1}^{N} r_n^2(S).
$$

By Lemma 5 (with high probability), we have that

$$
\begin{aligned}
R_N^2(S) &\leq N 4T^2 L_c'^2 \sum_{n=1}^{N} \beta_n^2 (1 + L_{\boldsymbol{h}}' + 2\beta_n L_{\boldsymbol{\sigma}}')^{2(T-1)} \left( \sum_{k=1}^{T} \|\boldsymbol{\sigma}_{n-1}(\boldsymbol{z}_{n,k})\| \right)^2 \\
&\leq N 4\beta_N^2 T^3 (1 + L_{\boldsymbol{h}}' + 2\beta_N L_{\boldsymbol{\sigma}}')^{2(T-1)} L_c'^2 \sum_{n=1}^{N} \sum_{k=1}^{T} \|\boldsymbol{\sigma}_{n-1}(\boldsymbol{z}_{n,k})\|^2.
\end{aligned}
$$

Taking the square root, we obtain

$$
R_N(S) \leq 2\beta_N T^{\frac{3}{2}}(1 + L_{\boldsymbol{h}}' + 2\beta_N L_{\boldsymbol{\sigma}}')^{T-1} L_c' \sqrt{N \sum_{n=1}^{N} \sum_{k=1}^{T} \|\boldsymbol{\sigma}_{n-1}(\boldsymbol{z}_{n,k})\|^2}.
$$

The result follows by noting that

$$
\sum_{n=1}^{N} \sum_{k=1}^{T} \|\boldsymbol{\sigma}_{n-1}(\boldsymbol{z}_{n,k})\|^2 \leq \tilde{\mathcal{I}}_N(\boldsymbol{h}^*, S).
$$

$\square$

### B.1.2 Bounding Model Complexities

The model complexity bounds from the continuous-time setting (c.f., Appendix A.2) extend seamlessly to the discrete-time setting. In the following we briefly sketch the bound for the oracle MSS.

**Lemma 19** (Discrete-time oracle model complexity). *For any $N \geq 1$,*

$$\tilde{\mathcal{I}}_N(\boldsymbol{h}^*, S^{ORA}) \leq \max_{\substack{\boldsymbol{\pi}_1,\ldots,\boldsymbol{\pi}_N \\ \boldsymbol{\pi}_n \in \Pi}} T \sum_{n=1}^{N} \left\| \boldsymbol{\sigma}_{n-1}\left(\boldsymbol{z}_n(t_{n,1})\right) \right\|^2. \tag{34}$$

*Proof.*

$$\tilde{\mathcal{I}}_N(\boldsymbol{h}^*, S^{\text{ORA}}) = \max_{\substack{\boldsymbol{\pi}_1,\ldots,\boldsymbol{\pi}_N \\ \boldsymbol{\pi}_n \in \Pi}} \sum_{n=1}^{N} \sum_{k=1}^{T} \left\| \boldsymbol{\sigma}_{n-1}\left(\boldsymbol{z}_{n,k}\right) \right\|^2$$

$$\leq \max_{\substack{\boldsymbol{\pi}_1,\ldots,\boldsymbol{\pi}_N \\ \boldsymbol{\pi}_n \in \Pi}} T \sum_{n=1}^{N} \max_{k \in \{1,\ldots,T\}} \left\| \boldsymbol{\sigma}_{n-1}\left(\boldsymbol{z}_{n,k}\right) \right\|^2$$

$$\leq \max_{\substack{\boldsymbol{\pi}_1,\ldots,\boldsymbol{\pi}_N \\ \boldsymbol{\pi}_n \in \Pi}} T \sum_{n=1}^{N} \left\| \boldsymbol{\sigma}_{n-1}\left(\boldsymbol{z}_n(t_{n,1})\right) \right\|^2.$$

$\square$

Note that the MSS $S$ can (but not has to) be restricted to the discretization of $[0,T]$ used for the dynamics and costs. The derivations for the equidistant and adaptive MSSs follow analogously, where we note that the bucket length $\Delta_n$ is to be defined with respect to a modified constant $\bar{\Gamma}_n$.

Finally, applying the measurement uncertainty bound presented in Appendix A.3, we obtain the following theorem.

**Theorem 20.** *Fix an arbitrary discretization $\boldsymbol{h}^*$ and assume that $\boldsymbol{h}^* \in \mathcal{H}_{n,B}^{d_x}$, the observation noise is i.i.d. $\mathcal{N}(\boldsymbol{0}; \sigma^2 \boldsymbol{I})$, and let $\|\cdot\|$ be the Euclidean norm. We model $\boldsymbol{h}^*$ with the GP model. The regret for the adaptive MSS is with probability at least $1 - \delta$ bounded by*

$$R_N(S^{ADP}) \leq \mathcal{O}\left( \beta_N T^2 (1 + L'_{\boldsymbol{h}} + 2\beta_N L'_{\boldsymbol{\sigma}})^{T-1} \sqrt{N \gamma_N} \right), \qquad m_n^{ADP} = \mathcal{O}(\beta_n).$$

## C Experimental Setup

### C.1 System's dynamics

Here we either give reference to the equations of the dynamical system or provide their equations:

- Cancer Treatment:

$$\dot{x}(t) = rx(t) \log\left(\frac{1}{x(t)}\right) - \delta u(t)x(t)$$

$$x(0) = 0.975, r = 0.3, \delta = 0.45$$

- Glucose in Blood:

$$\dot{x}_0(t) = -ax_0(t) - bx_1(t)$$
$$\dot{x}_1(t) = -cx_1(t) + u(t),$$
$$a = 1, b = 1, c = 1, x(0) = (0.75, 0)$$

- Pendulum:

$$\dot{x}_0(t) = x_1(t)$$
$$\dot{x}_1(t) = \frac{g}{l}\sin(x_0(t)) + u(t)$$
$$g = 9.81, l = 5.0$$

- Mountain Car:

$$\dot{x}_0(t) = 10x_1(t)$$
$$\dot{x}_1(t) = 3.5u(t) - 2.5\cos(3x_0(t))$$

- Cart Pole: Equation 6.1 and 6.2 of Kelly (2017)
- Furuta Pendulum: Equation 18 of Gäfvert (2016)
- Bicycle: Dynamics from (Singh and Theers, 2021)
- Quadrotor 2D: Equations 1, 2, 3 of Paing et al. (2020)
- Quadrotor 3D: Equations 12, 13, 14 of Thomas et al. (2017)

## C.2 System's tasks

In all considered settings the running cost has the following quadratic form:

$$c(\boldsymbol{x}, \boldsymbol{u}) = (\boldsymbol{x} - \boldsymbol{x}_T)^\top \boldsymbol{Q}(\boldsymbol{x} - \boldsymbol{x}_T) + (\boldsymbol{u} - \boldsymbol{u}_T)^\top \boldsymbol{R}(\boldsymbol{u} - \boldsymbol{u}_T).$$

Here, $\boldsymbol{x}_T$ is the target state, and $\boldsymbol{u}_T$ is the target control. Descriptively, in cancer treatment and glucose in Blood systems, our objective is to reduce the prevalence of unhealthy cells, while striving to be as minimally invasive as possible. For the pendulum, Furuta pendulum, and cart pole systems our goal is to skillfully swing the pendulum upwards from its lowest point. In the mountain car system, the challenge is to guide the car from the valley's depth to the hill's summit. Finally, in systems bicycle, quadrotor 2D, and quadrotor 3D our objective is to adeptly navigate the agent to reach a desired target state.

Table 3: Cost specifications for considered systems.

| System | Running $\boldsymbol{Q}$ | Running $\boldsymbol{R}$ | Target $\boldsymbol{x}_T$ | Target $\boldsymbol{u}_T$ |
|---|---|---|---|---|
| Cancer Treatment | $\boldsymbol{I}_1$ | $\boldsymbol{I}_1$ | $[0.54]$ | $\boldsymbol{0}_1$ |
| Glucose in Blood | $\boldsymbol{I}_2$ | $\boldsymbol{I}_1$ | $[0.47, 0.33]$ | $\boldsymbol{0}_1$ |
| Pendulum | $\boldsymbol{I}_2$ | $\boldsymbol{I}_1$ | $[0, 0]$ | $\boldsymbol{0}_1$ |
| Mountain Car | $\boldsymbol{I}_2$ | $\boldsymbol{I}_1$ | $[\frac{\pi}{6}, 0]$ | $\boldsymbol{0}_1$ |
| Cart Pole | $\boldsymbol{I}_4$ | $\boldsymbol{I}_1$ | $\boldsymbol{0}_4$ | $\boldsymbol{0}_1$ |
| Furuta Pendulum | $\boldsymbol{I}_4$ | $\boldsymbol{I}_1$ | $\boldsymbol{0}_4$ | $\boldsymbol{0}_1$ |
| Bicycle | $\boldsymbol{I}_4$ | $\boldsymbol{I}_2$ | $[2, 1, \frac{\pi}{6}, 0]$ | $\boldsymbol{0}_2$ |
| Quadrotor 2D | $\boldsymbol{I}_6$ | $10 \cdot \boldsymbol{I}_2$ | $\boldsymbol{0}_4$ | $[g \cdot m, 0]$ |
| Quadrotor 3D | $\mathrm{diag}([1, 1, 1, 1, 1, 1, 0.1,$ $0.1, 0.1, 1, 0.1, 0.1])$ | $\mathrm{diag}([5, 0.8, 0.8, 0.3])$ | $[0, 0, 0, 0, 0, 0,$ $0, 0, 1, 0, 0, 0]$ | $[0.15, 0, 0, 0]$ |

## C.3 Adaptive MSSs

In this section, we describe in detail the two adaptive MSSs from Section 5.

**Greedy Max Determinant** In the *Greedy Max Determinant* adaptive strategy we solve the following objective:

$$\max_{S \subset [0,T], |S| = M} \det\left(\boldsymbol{K}_S^n\right). \tag{35}$$

Here, $\boldsymbol{K}_S^n \stackrel{\text{def}}{=} [k_n(\widehat{\boldsymbol{z}}_n(t), \widehat{\boldsymbol{z}}_n(t'))]_{t,t' \in S}$ and $k_n(\boldsymbol{z}, \boldsymbol{z}') \stackrel{\text{def}}{=} k(\boldsymbol{z}, \boldsymbol{z}') - \boldsymbol{k}_n^\top(\boldsymbol{z})\left(\boldsymbol{K}_n + \sigma^2 \boldsymbol{I}\right)^{-1} \boldsymbol{k}_n(\boldsymbol{z}')$. The goal here is to find at every episode $n$ a set $S_n$ of $M$ time points from $[0, T]$. Solving problem (35) is NP-hard and we approximate it by solving it greedily:

---

**Algorithm 1** GREEDY MAX DETERMINANT

---

**Init:** $S_n = \{\mathrm{argmax}_{t \in [0,T]}\, k_n(\widehat{\boldsymbol{z}}_n(t), \widehat{\boldsymbol{z}}_n(t))\}$
**for** $k = 2, \ldots, M$ **do**

$$S_n = S_n \cup \left\{ \underset{t \in [0,T]}{\mathrm{argmax}}\, \det\left(\boldsymbol{K}_{S_n \cup \{t\}}^n\right) \right\}$$

---

**Greedy Max Kernel Distance**  The second adaptive MSS we consider in Section 5 is the *Greedy Max Kernel Distance*. Here we greedily solve the metric $M$-center problem in space $[0, T]$ with the kernel metric: $d_n(t, t')^2 \stackrel{\text{def}}{=} k_n(\widehat{\boldsymbol{z}}_n(t), \widehat{\boldsymbol{z}}_n(t)) + k_n(\widehat{\boldsymbol{z}}_n(t'), \widehat{\boldsymbol{z}}_n(t')) - 2k_n(\widehat{\boldsymbol{z}}_n(t), \widehat{\boldsymbol{z}}_n(t'))$ on the hallucinated trajectory.

---

**Algorithm 2** GREEDY MAX KERNEL DISTANCE

---

**Init:** $S_n = \{\text{argmax}_{t \in [0,T]} k_n(\widehat{\boldsymbol{z}}_n(t), \widehat{\boldsymbol{z}}_n(t))\}$
**for** $k = 2, \ldots, M$ **do**
$$S_n = S_n \cup \left\{ \underset{t \in [0,T]}{\text{argmax}} \, \underset{t' \in S_n}{\min} \, d_n(t, t') \right\}$$

---

### C.4 Episodes Hyperparameters

Table 4: We take a few tens of measurements per episode in each environment. We run the experiments for at most 40 episodes in each environment.

| System | Number of episodes $N$ | Number of Measurements per episodes $M$ | Time horizon $T$ |
|---|---|---|---|
| Cancer Treatment | 20 | 10 | 20 |
| Glucose in Blood | 20 | 10 | 0.45 |
| Pendulum | 20 | 10 | 10 |
| Mountain Car | 40 | 10 | 1 |
| Cart Pole | 40 | 10 | 10 |
| Furuta Pendulum | 40 | 10 | 10 |
| Bicycle | 20 | 10 | 10 |
| Quadrotor 2D | 20 | 10 | 8 |
| Quadrotor 3D | 25 | 20 | 15 |

### C.5 Practical Implementation

In practice, we approach solving the optimal control problem (2) with *offline planning* and *online tracking* strategy. Before the episode starts we obtain an open loop trajectory $\tilde{\boldsymbol{x}}_n(t), \tilde{\boldsymbol{u}}_n(t)$ by offline planning:

$$\tilde{\boldsymbol{x}}_n(t), \tilde{\boldsymbol{u}}_n(t) = \underset{\boldsymbol{x}(t), \boldsymbol{u}(t)}{\text{argmin}} \, \underset{\eta(t)}{\min} \int_0^T c(\boldsymbol{x}(t), \boldsymbol{u}(t)) \, dt$$
$$\text{s.t. } \dot{\boldsymbol{x}}(t) = \boldsymbol{\mu}_n(\boldsymbol{x}(t), \boldsymbol{u}(t)) + \beta_n \boldsymbol{\sigma}_n(\boldsymbol{x}(t), \boldsymbol{u}(t)) \boldsymbol{\eta}(t) \quad \text{(36)}$$
$$\boldsymbol{\eta}(t) \in [-1, 1]^{d_x}, \quad \forall t \in [0, T].$$

We solve the optimization problem (36) with Iterative Linear Quadratic Regulator (ILQR) (Li and Todorov, 2004). Then we track the open loop trajectory $\tilde{\boldsymbol{x}}_n(t)$ online with MPC:

$$\boldsymbol{u}(t) = \underset{\boldsymbol{u}(t), \boldsymbol{x}(t)}{\text{argmin}} \int_s^{s+T_{MPC}} \left( \|\boldsymbol{x}(t) - \tilde{\boldsymbol{x}}(t)\|_2^2 + \|\boldsymbol{u}(t) - \tilde{\boldsymbol{u}}(t)\|_2^2 \right) dt$$
$$\text{s.t. } \dot{\boldsymbol{x}}(t) = \boldsymbol{\mu}_n(\boldsymbol{x}(t), \boldsymbol{u}(t)) \quad \text{(37)}$$

Here, horizon $T_{MPC}$ depends on the system.

Table 5: The MPC horizon for considered systems.

| System | $T_{MPC}$ |
|---|---|
| Cancer Treatment | 5.0 |
| Glucose in Blood | 0.2 |
| Pendulum | 6.0 |
| Mountain Car | 1.0 |
| Cart Pole | 5.0 |
| Furuta Pendulum | 5.0 |
| Bicycle | 5.0 |
| Quadrotor 2D | 3.0 |
| Quadrotor 3D | 5.0 |

We solve MPC tracking problem (37) also with the ILQR.

# D   Additional Experiments on Number of Measurements

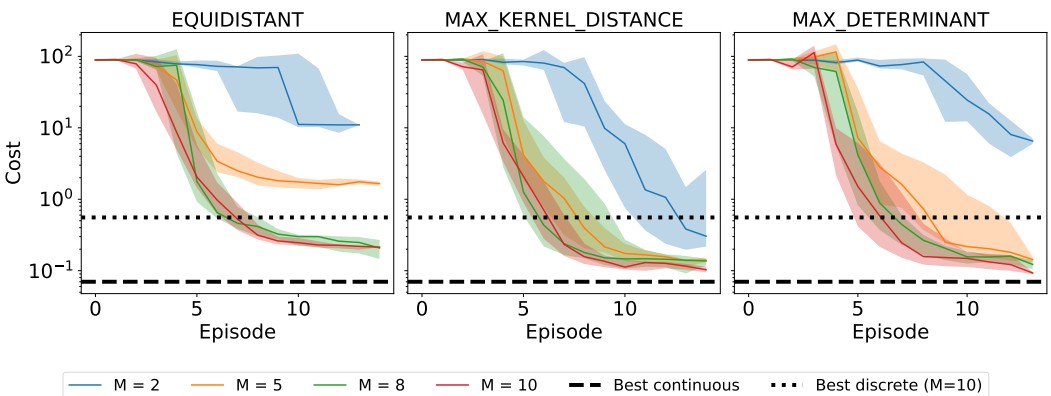

(a) We vary number of measurements we take per episode for the Pendulum system. As expected the more observation results in lower achieved cost for every MSS.

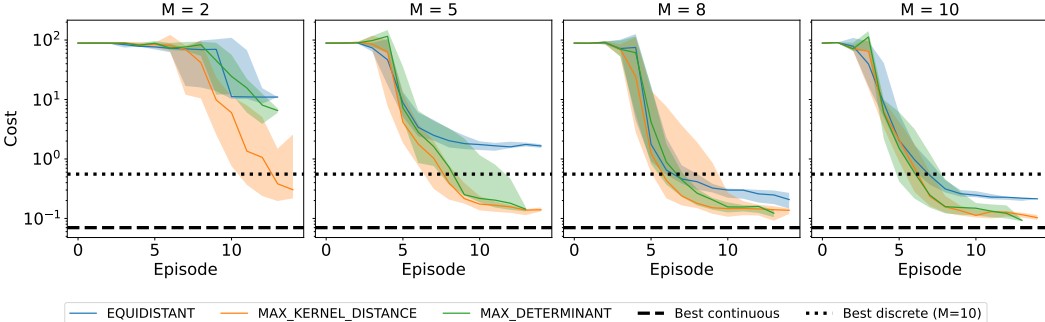

(b) For the same number of measurments we compare different MSSs. We see that equidistant MSS repeatedly suffers higher cost compare to the adaptive Max Kernel Distance and Max Deteriminant MSS.

