# OpenReview forum: "Efficient Exploration in Continuous-time Model-based Reinforcement Learning"
_NeurIPS.cc/2023/Conference — NeurIPS 2023 poster_

### Official Review · Reviewer_knGi · 2023-07-03

**Soundness:** 3 good
**Presentation:** 3 good
**Contribution:** 3 good
**Rating:** 5
**Confidence:** 3

**Summary:**

The submission considers online RL, while the true dynamics is continuous-time. To deal with this issue, the authors provide a continuous-time model-based method. The proposed method is special by
1. iteratively fitting ode-based models and deriving/learning the control from the fitted models
2. having the option to adaptively selecting the sampling timing.

The proposed method is able to well handle problems with continuous-time true dynamics, with both theoretical guarantees and empirical evidence.

**Strengths:**

The submission is novel. The proposed method incorporates continuous-time models with adaptive sampling, which to the best of my knowledge is the first.

The submission provides thorough results with both theoretical guarantees and also empirical experiments.

The studied problem is well motivated: there are problems where the data is generated in continuous-time in need of specialized methodology like the submission.

The writing is clear and easy to follow.

**Weaknesses:**

The submission is solid, and the adaptive MSS seems to be an important and useful technique. However, there is one concern bugging me, which seems a little crucial: it is not clear whether the continuous modeling is indeed needed.

My question is twofold: 1. whether it is necessary or practical to assume a continuous-time ground-truth dynamic; 2. whether it is necessary or helpful to use continuous-time models in the method for $f$

First of all, the continuous dynamics in section 2 and equation (1) may never be empirically achieved or used. There are very rare cases where one can really implement a policy in a continuous-manner as that equation. As a result, the policy functions considered will often time be piece-wise constant, which does not fit (1). It is not clear how this may affect the theory. When evaluating a policy, equation (1) may also need to be discretized. Therefore, in each episode, one can also fit a discrete-time model instead of a continuous-time one as suggested by the authors. I am wondering how the proposed method compares to this discrete-time version of the proposed method.

Second, the theoretical results like proposition 1 also holds for the discrete dynamics setting. The adaptive MSS may also be feasible in the discrete-time setting, which itself now is implemented in a discretized version.

Therefore, the observations above make me wonder whether the continuous-time part is really needed in the submission.

**Questions:**

1. In the experiments, are the cost function collected in a discrete-time manner or continuous-time manner? More specifically, are the results summation of costs or the integral of costs over time?

2. What difference might be, if we replace the ODE model by a discrete-time model, for example, like iLQR. Please demonstrate the differences both theoretically and empirically.

3. In experiments why does the proposed method with equidistant perform better than discrete zero-order hold OC? Are the two policy implemented using the same time interval on the same grid? Or does the proposed method calculates the integral over time of costs? Or does it use a different time interval than the competing discrete-time policy? What is the definition of the discrete zero-order hold OC?

4. Is it possible to implement MSS with a discrete-time assumed model? For example, I can assume that the true model is a discrete-time model which is just same as the Euler discretized version of the ODE in the submission, and then conduct adaptive MSS like in the submission. What is the problem of this naive ablation?

5. Would it be possible to explicitly list the dynamics of the experiments? The cost function is provided in the supplements but not the dynamics. I am concerned that the considered experiments are too simple like all linear ODEs, which may make the method not general enough.

As a summary, the submission definitely has novelty and contributions. However, I feel that the current method is not demonstrated clear enough, especially on what role the continuous-time modeling is playing in the method.

---

> ### Author Rebuttal · Authors · 2023-08-06
>
> Thank you for your comments and valuable feedback!
> ## Weaknesses
> 1. *Why continuous-time modeling:*
> Continuous time learning has several benefits over discrete-time modeling. For example, all systems in natural sciences are continuous in nature, therefore introducing priors into the learning problem is easier in continuous time (discrete-time priors are susceptible to the choice of discretization). Furthermore, when learning discrete-time dynamics, the sampling and control frequency is fixed to the discretization of the problem formulation. For a new choice of discretization, generally one has to relearn the dynamics. In continuous time, once the ODE is learned, the learned model can be used for any discretization.
> We appreciate the comment of the reviewer that real-world systems are controlled in discrete time and not continuously. In our analysis, we consider a general policy class. A special case of this policy class is piece-wise constant policy functions, which as the reviewer acknowledges are more practical (we do not explicitly include the time-dependency of the policy in eq (1) for simplicity). Furthermore, as we show in our results, we can obtain continuous-time performance with our algorithm empirically. Furthermore, we compare our algorithm with the true discrete-time models in Table 1 and show that our method outperforms the discrete-time case. Since we already outperform the ground truth discrete-time model, we did not evaluate a discrete-time model-based RL method additionally.
> In summary, continuous time modeling has several advantages over its discrete-time counterpart as we discuss above. Our problem formulation also considers the more practical class of piecewise constant policy functions, and in our results, we show that continuous time modeling achieves better performance. We hope we could convince the reviewer about continuous time modeling with our response.
> ## Questions
> 1. *Cost function evaluation:*
> The cost is always evaluated in the continuous-time setting. That means that no matter whether we control the system continuously or with a discrete zero-order hold, we evolve the dynamics in continuous time and compute the continuous-time cost (up to ODE solver precision).
> 2. *Difference between ODE vs discrete dynamics:*
> We are not sure if we understood the question exactly. Here is a potential response; we compare our continuous time modeling approach to the ground truth discrete-time model in Table 1. Our experiments show that our method outperforms the discrete-time case. Learning discrete-time dynamics is a well-studied problem, where similar theoretical guarantees exist [1].
> 3. *Equidistant performs better than discrete zero-order hold OC:*
> We refer the reviewer to the Author Rebuttal on how the costs are calculated in our experiments. As discussed in the authors rebuttal section discrete-time modeling and control are limited to the discretization schema. On the contrary continuous-time modeling benefits from representing the dynamics between the discretization nodes and adapting the controls accordingly. With equidistant MSS we take measurements equidistantly in time. However, we still learn a continuous time model and use it for continuous-time control.
> From Table 1, we see that the performance of our learned continuous-time controller is better than the discretized zero-order hold controller obtained using the true dynamics.
> 4. *MSS for discrete-time models:*
> We are not entirely sure we understood your question here. Here is a tentative answer:
> MSS can be used in the setting where we can only observe the state at some fixed discretized times. However, it still requires learning dynamics in continuous time. This is because discrete-time modeling requires observing transitions $(x_{k}, u_{k}, x_{k+1})$ at each time step. Did we provide the answer you had in mind or were you interested in something else?
> 5. *List of dynamics models:*
> All considered systems except for the Glucose in the blood system are not linear and standard in the literature. For the sake of completeness, we added them to the updated version of the paper.
>
> Having addressed all of the questions provided by the reviewer, and given the contributions of this paper, we would appreciate it if the reviewer would increase their score for our paper. We would be happy to answer any remaining questions or concerns.
> ## References
> [1] Curi, Sebastian, Felix Berkenkamp, and Andreas Krause. "Efficient model-based reinforcement learning through optimistic policy search and planning." Advances in Neural Information Processing Systems 33 (2020).

---

> > ### Author Response · Authors · 2023-08-17
> > **Follow up on rebuttal**
> >
> > We hope we could address your concerns adeptly. We would further like to emphasize another benefit of continuous-time modeling over its discrete-time counterpart.
> >
> > Discrete-time algorithms such as H-UCRL only work for the setting where the measurement frequency and control frequency are the same. Continuous time modeling can separate these two elements. We leverage this in our work, by proposing MSS that only collects data that benefits in learning the ODE.
> >
> > In summary, due to continuous-time modeling, our proposed method has more control over the system’s measurement and control frequencies, incorporates a general class of policies such as piece-wise constant policies with any choice of control frequency, and *outperforms the true discrete-time model*.
> >
> > We have updated the paper to further discuss the benefits of continuous-time modeling. It is much appreciated if you could please reconsider your assessment, or respond with questions/suggestions so that we can improve the paper in this regard.

---

### Official Review · Reviewer_xqxW · 2023-07-06

**Soundness:** 3 good
**Presentation:** 3 good
**Contribution:** 3 good
**Rating:** 7
**Confidence:** 3

**Summary:**

This paper proposes a continuous time framework for model based reinforcement learning. Their algorithm OCoRL solves the optimal control problem eq (1) by: 1, selecting optimistic policy. 2, rollout to collect data. 3, update model estimation and statistics. Specifically, they study the measurement selection strategy (sampling state-action in the continuous time framework), and show both theoretically and empirically the effect of different measurement selection strategy.


**Strengths:**

The paper models the optimal control or reinforcement learning problem in continuous time (eq (1)), which is elegant and also convenient for algorithm design and analysis. The realization of state-action at discrete time steps is naturally modeled as discrete-time sampling from the continuous time trajectory (the measurement selection strategy (MSS)). The proposed algorithm alternates between: optimistic planning given current statistical model and data collection for a more accurate model. They further proposed the adaptive MSS that samples state-action time points based variance of model estimate in the state-action space (rollout trajectory).

Both theoretical and empirical analysis are provided and showcase the impact of different discrete sampling strategies.


**Weaknesses:**

The algorithm enjoys elegance and nice theoretical properties, but it can be difficult to realize in practice. Especially, the first step in OCoRL that solves optimistic policy can be very time-consuming in practice. The authors mentioned in Appendix C that it is approximately solved by Iterative Linear Quadratic Regulator, but still it can be impractical.

The experiments mainly focus on investigating the impact of measurement selection strategies. A comparison in terms of performance: time/computation complexity and over all cost with existing approach (PPO, etc) would better help readers evaluate this approach

**Questions:**

See weakness

**Limitations:**

See weakness

---

> ### Author Rebuttal · Authors · 2023-08-06
>
> Thank you for your positive and valuable feedback!
>
> # Weaknesses and Questions
>
> 1. *A comparison in terms of performance: time/computation complexity and over all cost with existing approach (PPO, etc) would better help readers evaluate this approach*:
>  As you correctly notice the first step of OCORL, namely solving optimization problem (2) from the paper is challenging. We tried to solve optimzation problem (2) using black box optimal control optimizers. In particular, we used ILQR coupled with MPC. We solved the continuous-time optimal control problem (2) by discretizing it with a small time step such that the change in the achieved cost was negligible with coarser discretization.
> In Table 1, we compare our work with the ground truth continuous-time dynamics and its discretization. The baselines suggested by the reviewer (PPO etc.) operate in discrete time. Since we already compare our algorithm to the ground-truth discrete-time system, we did not include other discrete-time baselines. We would also like to highlight that the number of collected transitions is only in the range of [200, 500], this is a considerable gain in sample efficiency compared to model-free methods such as SAC or PPO.

---

### Official Review · Reviewer_7bYe · 2023-07-06

**Soundness:** 3 good
**Presentation:** 3 good
**Contribution:** 3 good
**Rating:** 6
**Confidence:** 4

**Summary:**

This paper proposes a continuous-time model-based reinforcement learning method for controlling fully observed dynamical system environments where there is a cost to take a sample of the state. A Gaussian process dynamics model is used, and a novel adaptive measurement selection strategy is proposed to determine when to take a sample of the state, such that the overall optimization converges to the optimal policy in the limit of infinite trials. The proposed method (OCoRL) is theoretically analyzed to show a general regret bound holds for any measurement selection strategy and is empirically verified across a range of dynamical system environments.

**Strengths:**

* The proposed method OCoRL and associated general regrets bound that holds for any measurement selection strategy appear novel.
* The paper is well-written and clearly laid out.
* The no-regret algorithm for nonlinear dynamical systems in the continuous-time RL setting seems widely applicable and relevant to the RL and ML community.
* The code is reproducible and easily extendable, being well documented throughout.

**Weaknesses:**

* How does the proposed method perform when the state differential is not observed $\dot{x}_n(t)$ and has to be inferred? Could you perform an ablation of this to show that the proposed method is still practical?
* Why is the noise only added to the observed state derivatives $\dot{x}_n(t)$? Perhaps it could be more realistic to consider noise added to both the observed state $x_n(t)$ and the observed state derivative $\dot{x}_n(t)$?
* Line 118: "predicted mean and epistemic uncertainty". How does the model guarantee that you only measure the "epistemic uncertainty" and not the "epistemic uncertainty and aleatoric uncertainty"? This was not clear, and I presume the model learns both. If so, how can you split the uncertainty to only use the epistemic uncertainty, as outlined in the method?
* The adaptive MSS assumes that $m_n=\left \lceil{T/ \Delta_n}\right \rceil$, i.e., that a sample must be taken in each uniform interval $\Delta_n$ of time. This seems overly restrictive. Can OCoRL be adapted to skip taking a sample in some $\Delta_n$ of time, i.e., where they are not needed or informative?
  * On this, can a further ablation be performed where $m_n$ is varied across all the environments and baselines to empirically verify the adaptive MSS claims for wide ranges of $m_n$?
* Unsubstantiated claim of "We compare the adaptive and equidistant MSSs on all systems and observe that the adaptive MSSs consistently perform better than the equidistant MSS". Table 1, shows that adaptive MSSs can perform better than equidistant MSS only in certain environments and that equidistant MSS achieves the same performance (final cost) within error bounds for the environments of Cancer Treatment, Pendulum, and Mountain Car.

**Questions:**

* Could this approach be generalized to other environments that are not implicitly defined by an ODE, such as other types of differential equations, e.g., delayed differential equations, stochastic differential equations, etc? Furthermore, can this approach work for partially observable environments?
* What is the trade-off of varying the number of measurements taken, i.e., an ablation of varying M for all experiments? Does the algorithm still hold under these settings? Can this be demonstrated empirically?
* Could the OCoRL method be benchmarked against the closest related works of Yildiz et al. (2021) and Du et al. (2020)?
* (From above): Can OCoRL be adapted to skip taking a sample in some $\Delta_n$ of time, i.e., where they are not needed or informative?
* Typo: Line 339: "Figure 1" -> "Figure 2".

**Limitations:**

The limitations are addressed with the assumptions outlined in Section 2.1.

---

> ### Author Rebuttal · Authors · 2023-08-06
>
> Thanks a lot for the positive and valuable feedback!
> ## Weaknesses
> 1. *Case when state derivatives $\\dot{x}(t)$ are not observed:*
> We thank the reviewer for this very interesting question. When the state derivatives are not observed one can apply several techniques to obtain them (e.g., using finite differences, interpolation methods, etc, [1-3]). We consider the incorporation of the derivative estimation inside the RL loop as an interesting avenue for future work.
> 2. *Noise also in the state $x(t)$:*
> The noise can be added to the states $x(t)$ as well (only as a measurement noise though). However, in the analysis, we would then need to transfer the noise from the inputs to the outputs via Taylor approximation and also update the $\sigma$ of the sub-Gaussian noise assumption. For simplicity and disposition flow we assumed noise only on the outputs of learned function, i.e. $\dot{x}(t)$.
> 3. *Measuring uncertainty:*
> Yes, the model learns both, aleatoric and epistemic uncertainty. We split noise into aleatoric and epistemic parts and use only epistemic part for planning. In the GP case, we learn homoscedastic aleatoric uncertainty by optimizing noise variance $\sigma^2_{ale}(x) = \sigma^2$ in the negative log-likelihood term: $\frac{1}{2}\dot{\mathbf{y}}^\top (\mathbf{K} + \sigma^2I)^{-1}\dot{\mathbf{y}} + \frac{1}{2}\text{logdet}(\mathbf{K} + \sigma^2I)$, the epistemic uncertainty (epistemic variance) we obtain from the formula: $\sigma^2_{epi}(x) = k(x, x) - k(x, \mathbf{X})(\mathbf{K} + \sigma^2I)^{-1}k(\mathbf{X}, x) $. Here we denoted by $\mathbf{K}$ the covariance matrix built from the observations, and $k(x, \mathbf{X})$ is a vector of kernel evaluations between point $x$ and all observations. When we use deep ensembles for modeling dynamics, every member of the ensemble learns to predict mean and aleatoric (heteroscedastic) uncertainty (variance). To obtain (approximate) epistemic standard deviation, we take, as is commonly done, a standard deviation of the predicted means of the ensemble.
> 4. *Skipping measurement in* $\Delta_n$ *interval time*:
> With the assumptions of the paper, we were not able to get rid of sampling in every interval $\Delta_n$ of time. However, if we for example assume some kind of stability (so that the trajectory doesn't have the option to deviate exponentially fast in time) we can alleviate this dependence. The second possible approach is to consider event-triggered sampling, where we would sample (take a measurement) only when the epistemic uncertainty at the true state would surpass a certain value. Again, for this kind of approach, we would need another assumption, namely to be able to continuously monitor the system (which one could argue can be done with hardware). Regarding changing the $m_n$ in our experiments: If we let $m_n$ be large, the difference between all MSS would become negligible. The main difference between the different MSSs arises when the number of collected data samples is small. We chose $m_n$ small enough so that the difference is visible. However, upon the reviewer's request, we performed an ablation study for different values of $m_n$ on the pendulum environment. We have attached a figure in the Author rebuttal section with our results. It is visible from the figure that for small values of $m_n$ there is a significant difference between the MSSs, while for larger MSSs this difference vanishes.
> 5. *Unsubstantiated claim based on Table 1:*
> Thanks for spotting that, we have corrected the claim in the revised version. From the table, it is only possible to claim that adaptive MSS outperforms equidistant MSS on the Cancer Treatment, Pendulum, Bicycle, Furuta Pendulum, Quadrotor 2D, and Quadrotor 3D environments while performing on par (within the overlapping confidence sets) on Glucose in Blood, Mountain Car, and Cart Pole environments.
> ## Questions
> 1. *Generalization to other environments:*
> In this work, we only consider systems driven by an ODE. Other types of systems like delayed differential equations, stochastic differential equations, and partially observable environments are still an open problem and we leave them as exciting future work.
> 2. *Varying number of taken measurements M:*
> With a small M (small number of observations) adaptive MSS performs much better compared to equidistant MSS. When the number of measurements grows the adaptive and equidistant MSS should result in similar performance (on our experiments) since we collect enough data in all regions (important and unimportant). We refer the reviewer to the newly added figure above for an ablation study.
> 3. *Comparison to Yildiz et al. (2021) and Du et al. (2020):*
> The setting of Yildiz et al. (2021) and Du et al. (2020) is a bit different since they don't consider access to noisy derivatives as we do. To compare with the work of Yildiz et al. (2021) we run experiments with a mean (greedy) planner (this is the planner used by Yildiz et al (2021)) instead of our optimistic one. As we can see in Figure 3 in the paper the optimistic planner is faster at finding the optimal policy.
> 4. *Skipping observations:*
> In practice (experimentally), we can indeed skip taking the measurements in some $\Delta_n$ interval of time, if the uncertainty there is not high (in hallucination), however with this we lose theoretical guarantees.
> 5. *Typo:*
> Thanks for spotting that, we were indeed referencing Figure 1. We updated the typo in the updated version of the paper
> ## References
> [1] Treven, L., Wenk, P., Dorfler, F., and Krause, A. (2021). Distributional gradient matching for learning uncertain neural dynamics models. Advances in Neural Information Processing Systems.
>
> [2] Chartrand, R. (2011). Numerical differentiation of noisy, nonsmooth data. International Scholarly Research Notices, 2011.
>
> [3] Knowles, I. and Renka, R. J. (2014). Methods for numerical differentiation of noisy data. Electron. J. Differ. Equ, 21:235–246.

---

> > ### Comment · Reviewer_7bYe · 2023-08-18
> >
> > Thank you for your detailed responses. I see your point about the restrictive assumption of taking a sample in every $\Delta_n$ time interval. I am keeping my original score.

---

### Official Review · Reviewer_ffit · 2023-07-11

**Soundness:** 3 good
**Presentation:** 2 fair
**Contribution:** 2 fair
**Rating:** 5
**Confidence:** 3

**Summary:**

This paper introduces a novel algorithm for efficient exploration in continuous-time model-based reinforcement learning. The algorithm represents continuous-time dynamics using nonlinear ODEs and captures epistemic uncertainty using probabilistic models. The analysis shows that the approach achieves sublinear regret with significantly fewer samples, making it a promising solution for various applications. The paper also develops a practical adaptive measurement selection strategy that reduces the number of measurements per episode while retaining regret guarantees. The benefits of continuous-time modeling and planning with optimism are demonstrated in several environments. The authors aim to catalyze further exploration within the RL community regarding the potential of modeling dynamics in a continuous-time framework.

**Strengths:**

This paper appears to be one of the pioneering works in the field of continuous-time reinforcement learning (RL) applied to nonlinear dynamical systems.

Furthermore, the utilization of epistemic uncertainty for measurement selection, a technique commonly employed in active learning but rarely explored in RL, adds a unique and valuable dimension to this study.

In addition to these notable contributions, the paper introduces several innovative techniques and investigates their efficacy across various experimental environments.


**Weaknesses:**

The paper encompasses a wide range of techniques, including model-based RL, continuous-time RL, and aperiodic strategies. However, the abundance of new content may make it challenging to write and comprehend the relationship between each technique and the overall idea. To enhance clarity and understanding, it would be beneficial to reorganize the paper and provide clearer connections between the different techniques employed.

Regarding Table 1, it appears that, under a known true model, discrete-time control outperforms continuous-time OC without the inclusion of MSS. Consequently, it may not be fair to directly compare continuous-time OC with MSS and discrete-time control since the conditions and components differ significantly between these approaches. A more appropriate comparison could be made between continuous-time OC and discrete-time control without MSS.

To provide a more comprehensive evaluation, it is advisable to compare this method with a broader range of other proposed RL methods. The current results may not fully demonstrate the performance of OCoRL, and including additional comparisons would enhance our understanding of its capabilities.

In Figure 3, it would be beneficial to consider incorporating more environments for experimentation. Relying solely on a single environment may not provide enough evidence to draw robust conclusions. Including a broader set of environments would strengthen the validity and generalizability of the findings.


**Questions:**

In which scenarios are measurements considered costly? Have you discussed in the Introduction whether there are enough scenes that truly require continuous-time systems due to costly measurements?

In the problem setting, the computation of cumulative regret relies on an optimal policy. However, if the optimal policy is unattainable for a particular problem, can your method still be applied? Please provide an explanation.

Regarding Line 122, is it possible for re-calibration techniques to provide accurate uncertainty estimation while maintaining a low time complexity?

---

> ### Author Rebuttal · Authors · 2023-08-06
>
> First, thank you a lot for your positive and valuable feedback! We will indeed incorporate the proposed ideas in the updated version of the paper.
>
> ## Weaknesses
>
> 1. *Enhance Clarity:*
> We thank the reviewer for the feedback. We have added the following summary of our method at the end of section 3:
> *OCORL consists of two key and orthogonal components; (i) optimistic policy selection and (ii) measurement selection strategies (MSSs). In optimistic policy selection, we optimistically, w.r.t. plausible dynamics, plan a trajectory and rollout the resulting policy. We study different MSSs, such as the typical equidistant MSS, for data collection within the framework of continuous time modeling. Furthermore, we propose an adaptive MSS that measures data where we have the highest uncertainty on the planned (hallucinated) trajectory. We show that OCORL suffers no regret for the equidistant, adaptive, and oracle MSSs.*
>
> 2. *Comparisons in Table 1:*
> Note that the numbers in the table represent the cost, so the smaller the better. Hence continuous time OC under the true model is the best (lowest) value any controller can achieve (up to numerical calculations). Since in the discrete-time control setting, we are only allowed to change the controller at equidistant times, the cost (computed on the realized continuous trajectory) is strictly larger (worse).
>
> 3. *Comparison to other methods:*
> We could also compare it to any other discrete-time algorithm SAC, PPO, etc., but since our algorithm controls the systems continuously, it outperforms even the best possible discrete-time control problem (i.e., the case when we know the true dynamics and can solve the optimal control problem with any solver) we decided to show the comparison only to the best possible control in discrete time (with given discretization).
>
> 4. *Further experiments on optimism vs greedy exploration*
> The comparison between greedy and optimistic planning has been done in several related works, e.g. in [1], where they show that optimistic planning performs better than greedy planning, especially in environments with scarce rewards or large action penalties. Since this is a well-studied problem and not within the focus of our paper we decided not to add any additional environments for the demonstration due to space restrictions. If the reviewer still insists on other environments we can add them to the final version.
>
>
>  ## Questions:
> 1. *Costly Measurements:*
> There is plenty of scenarios when taking measurements is costly, in the introduction we consider a case when a patient is coming to the doctor for medical checks. We don't want him to come too often (every appointment at the doctor is very costly), but only when it is necessary (the uncertainty about the development of the disease is large). Another instance can be found in wireless control systems, where there are constraints on energy, computation, and communication capacity. In such systems, communication should take place only when there's pertinent information to share [2].
>
> 2. *Unattainable optimal policy:*
> Not sure that we understand the second question exactly, please let us know if the following is what you meant: It can happen that the solution to the optimal control problem is not attainable (in a sense of inf, but the minimum is not attained). For the analysis, we assume the minimum is attained. Just for running the algorithm, we don't need that assumption.
>
> 3. *Recalibration techniques:*
> When Modelling dynamics with GPs we don't need to recalibrate the model since GPs provide statistically sound confidence sets by design (with the right value of $\beta_n$). For the recalibration of an Ensemble of Neural Networks, we need to solve $d_x$ (output of a neural network) optimization problem over a scalar variable, which can be solved very fast in practice, in our experiments recalibration usually takes only around 10ms.
>
> ## References
>
> [1] Curi, S., Berkenkamp, F., and Krause, A. (2020). Efficient model-based reinforcement learning through optimistic policy search and planning. Advances in Neural Information Processing Systems, 33:14156–14170.
>
> [2] Anta, Adolfo, and Paulo Tabuada. "To sample or not to sample: Self-triggered control for nonlinear systems." IEEE Transactions on automatic control 55.9 (2010): 2030-2042.

---

> > ### Comment · Reviewer_ffit · 2023-08-21
> >
> > Thank you for your rebuttal. Considering your rebuttal and comments of other reviewer, I stick to the current score.

---

### Author Rebuttal · Authors · 2023-08-09

We thank all the reviewers for their valuable and useful feedback. We believe there is been some confusion around the discrete-time control setting we consider in our work. Accordingly, we have clarified this further in the appendix of the updated paper. We include a summary below:

1. When we learn a system that follows the ODE $\dot{x} = f^*(x, u)$ in discrete time, we can only learn the discretized dynamics of the system, i.e., $x_{k+1} = f(x_k, u_k)$. This approach is severely limited to the choice of discretization.
2. For the learned discretized system $x_{k+1} = f(x_k, u_k)$, we *cannot model (predict) the behavior of the system between two time steps.* Therefore, our control inputs can only be changed at the discretization frequency.
3. Nonetheless, the underlying system is still evolving continuously over time. Hence, even if we model and control the system with the discrete model, we evaluate the state evolution and the corresponding cost with an ODE integrator, where the control is changing at the discretization frequency.

Lastly, we have attached a pdf of an additional experiment that studies different measurement selection strategies as requested by the reviewers. We included the results of the additional experiment in the appendix of the updated version of the paper. We would be happy to further update the paper if there are any remaining questions or feedback from the reviewers.

---

### Decision · Program_Chairs · 2023-09-21

**Decision:**

Accept (poster)

**Comment:**

The discussion is inclined towards acceptance.


We encourage the authors to consider the feedback provided in the review process.